



# A Hydrological Prediction System Based on the SVS Land-Surface Scheme: Implementation and Evaluation of the GEM-Hydro platform on the watershed of Lake Ontario

Étienne Gaborit[1,*], Vincent Fortin[1], Xiaoyong Xu[2], Frank Seglenieks[3], Bryan Tolson[2], Lauren M. Fry[4], Tim Hunter[5], François Anctil[6], and Andrew D. Gronewold[5]

[1]Environment Canada, Environmental Numerical Prediction Research (E-NPR), Dorval, H9P1J3, Canada.
[2]University of Waterloo, Civil and Environmental Engineering Dpt., Waterloo, N2L3G1, Canada.
[3]Environment Canada, Boundary Water Issues, Burlington, L7S1A1, Canada.
[4]U.S. Army Corps of Engineers, Detroit District, Great Lakes Hydraulics and Hydrology Office, Detroit, 48226, U.S.A.
[5]NOAA Great Lakes Environmental Research Laboratory (GLERL), Ann Arbor, 48108, U.S.A.
[6]Civil and Water Engineering department, Université Laval, Québec, G1V0A6, Canada.

*Correspondence to*: Étienne Gaborit (Etienne.Gaborit@Canada.ca)

**Abstract.** This work describes the implementation of the distributed GEM-Hydro runoff modeling platform, developed at Environment and Climate Change Canada (ECCC) over the last decade. The latest version of GEM-Hydro combines the SVS (Soil, Vegetation and Snow) land-surface scheme and the WATROUTE routing scheme in order to provide streamflow predictions on a gridded river network. SVS is designed to be two-way coupled to the GEM (Global Environmental Multi-scale) atmospheric model exploited by ECCC for operational weather and environmental forecasting. Although SVS has been shown to accurately track soil moisture during the warm season, it has never been evaluated before for hydrological prediction. This paper presents a first evaluation of its ability to simulate streamflow for all major rivers flowing into Lake Ontario. The skill level of GEM-Hydro is assessed by comparing the quality of simulated flows to that of two established hydrological models, MESH and WATFLOOD, which share the same routing scheme (WATROUTE) but rely on different land-surface schemes. All models are calibrated using the same meteorological forcings, objective function, calibration algorithm, and watershed delineation. Results show that GEM-Hydro performs well and is competitive with MESH and WATFLOOD. A computationally efficient strategy is proposed to calibrate the land-surface model of GEM-Hydro: a simple unit hydrograph is used for routing instead of its standard distributed routing component. The distributed routing part of the model can then be run in a second step to estimate streamflow everywhere inside the domain. Global and local calibration strategies are compared in order to estimate runoff for ungauged portions of the Lake Ontario watershed. Overall, streamflow predictions obtained using a global calibration strategy, in which a single parameter set is identified for the whole watershed of Lake Ontario, show skills comparable to the predictions based on local calibration. Hence, global calibration provides spatially consistent parameter values, robust performance at gauged locations, and reduces the complexity and computational burden of the calibration procedure. This work contributes to the Great Lakes Runoff Inter-



comparison Project for Lake Ontario (GRIP-O) which aims at improving Lake Ontario basin runoff simulations by comparing different models using the same input forcings.

**Key words.** Distributed models, GEM-Hydro, Local and global calibrations, Ungauged catchments, Unit hydrograph.

## Introduction

Given the continuous increase in precipitation forecast skill of Numerical Weather Prediction (NWP) systems, as documented for example over the United States (US) by Sukovich et al. (2014), it is becoming possible to obtain skillful runoff forecasts directly from NWP model outputs, and streamflow forecasts by routing these gridded runoff fields. Indeed, modern NWP models all simulate to some extent the snow, vegetation, and soil processes that contribute to the generation of runoff and streamflow. The Global Flood Awareness System (GloFAS) operates in such a fashion (Alfieri et al., 2013) to

issue global streamflow anomaly forecasts on a low-resolution (0.1 degree) grid. Going beyond anomaly forecasts (which are bias corrected based on a model climatology) to obtain unbiased short-term streamflow forecasts is more challenging due to limitations of operational Land-Surface Schemes (LSS), which are generally geared towards improving weather forecasts, sometimes at the cost of not representing (or misrepresenting) surface and subsurface hydrological processes that are critical to hydrological simulation. Many of these limitations are documented in Clark et al. (2015) and Davison et al. (2016).

Hydrological processes in land-surface models used for NWP are improving quickly (Balsamo et al., 2009; Masson et al., 2013; Alavi et al., 2016; Wagner et al., 2016), as soil water content and snow are recognized as important sources of their predictability that remain to be fully tapped into (Koster et al., 2004; Entekhabi et al., 2010). Environment and Climate Change Canada (ECCC), the Canadian department that provides operational weather and environmental forecasts, is in the process of implementing a major upgrade to the LSS used by its NWP model, the Global Environmental Multi-scale model

(GEM). This new scheme, named SVS for Soil, Vegetation and Snow, has been devised in order to assimilate space-based soil moisture retrievals as well as surface data, and has proven efficient at simulating soil moisture and brightness temperature (Alavi et al., 2016; Husain et al., 2016). SVS will replace the Canadian version of the ISBA scheme (*Interaction Sol-Biosphère-Atmosphère*) that has been used operationally since 2001 (Bélair et al., 2003). This paper presents the first evaluation of the capabilities of the new SVS scheme for hydrological prediction in Canada, focusing on the tributaries of

Lake Ontario.

GEM's LSSs can be run either two-way coupled to the atmospheric model or offline, using GEM or other observed atmospheric forcing. The platform for running GEM offline is known as GEM-Surf (Bernier et al., 2011). Runoff obtained from the LSS can then be routed to the outlet of the watershed using the WATROUTE routing scheme (Kouwen, 2010). This configuration is known as GEM-Hydro.

Our current evaluation of GEM-Hydro focuses on the Lake Ontario watershed for many reasons including (1) the socio-economic impacts that improvements to streamflow and lake level prediction skill can have on a region of Canada that is quite populated and industrialized; (2) the large amount of data available for model set up, calibration, and validation,



compared to other regions of Canada; and (3) the fact that this is a Canada-USA transboundary watershed which is co-managed by ECCC and US Army Corps of Engineers (USACE) staff, in accordance with water level management rules set by the International Joint Commission (IJC) for each control structure, including the Moses-Saunders power dam at Cornwall, the outlet of Lake Ontario (Fig. 1).

Different cascades of interconnected models have been developed over the years to simulate the Great Lakes water levels and thermodynamics, such as Wiley et al. (2010), Deacu et al. (2012), and Gronewold et al. (2011), the latter describing the Advanced Hydrologic Prediction System (AHPS), a seasonal water supply and water level forecasting system developed by the National Oceanic and Atmospheric Administration (NOAA) Great Lakes Environmental Research Laboratory (GLERL) in the mid-1990s that has since been employed operationally (with few changes in methodology) by the USACE and regional hydropower authorities. Recently, ECCC has implemented a short-term (84-h) operational water cycle prediction system for the Great Lakes and St. Lawrence River (WCPS-GLS) that uses coupled atmospheric, hydrologic, and hydrodynamic models (Durnford et al., in preparation). This system makes use of the same platform used in this study, GEM-Hydro, but relies on the simpler ISBA LSS.

To our knowledge, GLERL's AHPS and ECCC's WCPS-GLS systems are the only two systems that provide inflow forecasts for each of the Great Lakes on both sides of the Canada-US border, and neither relies on very sophisticated hydrological models. The need for improving simulations and forecasts of runoff to the Great Lakes is recognized by both agencies (Gronewold and Fortin, 2012). Multiple additional hydrologic models are indeed available (Coon et al., 2011), however their spatial domains are typically constrained to either the US or Canada. Before embarking on an upgrade of operational systems, GLERL and ECCC agreed to perform a number of intercomparison studies under the umbrella of the Great Lakes Runoff Intercomparison Project (GRIP), in order to better understand the status of existing systems, and to set a benchmark for model performance against which future models could be compared. The first study was conducted on the Lake Michigan (GRIP-M) watershed by Fry et al. (2014) who compared historical runoff simulations from dissimilar hydrologic models using different calibration frameworks and input data. Amongst the models compared were GLERL's Large Basin Runoff Model (LBRM; Croley and He, 2002) that is part of GLERL's AHPS, the NOAA National Weather Service model (NWS; Burnash, 1995), and ECCC's MESH distributed model (*Modélisation Environnementale* – Surface and Hydrology; Pietroniro et al., 2007; Haghnegahdar et al., 2014). A second configuration of MESH was also included, based on Deacu et al. (2012), from which evolved the configuration of GEM-Hydro used by Durnford et al. (in preparation) for the operational WCPS-GLS system. The NWS model performed best in terms of Nash-Sutcliffe skill, but was positively biased, perhaps because of its typical use as a flood forecasting tool. Overall, it was difficult to attribute any difference in model results to the model structure, given that different forcing data and calibration procedures had been used by each contributor to the project.

The GRIP project was extended next to Lake Ontario (GRIP-O) by Gaborit et al. (in Press), who compared two lumped models, namely LBRM and GR4J (*modèle du Génie Rural à 4 paramètres Journalier*; Perrin et al., 2003), with the exact same forcing data and calibration framework. Two precipitation datasets were used as input: the Canadian



Precipitation Analysis (CaPA; Lespinas et al., 2015), and a Thiessen polygon interpolation of the Global Historical Climatology Network - Daily (GHCND; Menne et al., 2012). CaPA is a near real-time quantitative precipitation estimate product from ECCC that is available on a 10-km grid for all of North America:

([http://collaboration.cmc.ec.gc.ca/cmc/cmoi/product_guide/submenus/capa_e.html](http://collaboration.cmc.ec.gc.ca/cmc/cmoi/product_guide/submenus/capa_e.html)).

The main findings of the first GRIP-O study are that the performance of the models was very satisfactory, whatever the precipitation database used, for all tributaries of Lake Ontario, despite the fact that most tributaries have a regulated flow regime. This satisfactory performance justifies the use of CaPA as a precipitation forcing dataset in later studies, especially for distributed models which require gridded precipitation as input. The performance of lumped models also provides a reference level of performance when evaluating distributed hydrological models.

The present work is an extension of the first GRIP-O study but focused on distributed hydrological models. Distributed models are more complicated to implement and more computationally-intensive than lumped ones, but have a broader range of applications. The main objective of this study is to propose a methodology for calibrating the distributed GEM-Hydro platform developed by ECCC in order to improve streamflow simulations for Lake Ontario, which we expect would ultimately propagate into improved simulations of Lake Ontario Net Basin Supplies (or NBS, the sum of lake

tributary runoff, overlake precipitation, and overlake evaporation: Brinkmann 1983). A second objective is to compare GEM-Hydro with two other distributed models (inter-comparison study) in order to identify avenues to further improve GEM-Hydro. And a third objective is to propose and evaluate a method for estimating runoff for the ungauged parts of the watershed.

## 1 Methodology

**1.1 Models**

Three different platforms are compared in this study: MESH, WATFLOOD, and GEM-Hydro. They have in common a distributed representation of most hydrological processes occurring in a watershed and a structure organized around two main components: a LSS for the representation of surface processes (evapotranspiration, infiltration, snow processes, water circulation in the soils), and a river routing scheme for simulating water transport in the streams, which

consists of WATROUTE for all models. WATROUTE is a 1-D hydraulic model relying mainly on flow directions and elevation data (Kouwen 2010). It routes to the catchment outlet the surface runoff and recharge produced by the surface schemes. In WATROUTE, runoff directly feeds the streams while recharge can be provided to an optional Lower Zone Storage (LZS) compartment, representing superficial aquifers, which releases water to the streams. WATFLOOD and GEM-Hydro make use of the LZS, whereas recharge from MESH feeds directly into the stream.

The version of MESH used in this study relies on version 3.6 of the Canadian LAnd Surface Scheme (CLASS). Each grid cell is subdivided in a number of tiles, and each tile is classified as belonging to one of a number of grouped



response unit (GRU). Each GRU has an associated parameter set which needs to be calibrated. In this paper, we follow the calibration strategy advocated by Haghnegahdar et al. (2014) for MESH.

GEM-Hydro is very similar to MESH, but is tied to the LSSs available in GEM: ISBA and SVS. A previous study on the same watershed demonstrated the clear superiority of SVS over ISBA, especially in regard to the baseflow component of the streamflow (see Gaborit et al., 2016). We thus only use SVS with GEM-Hydro in this paper.

WATFLOOD (Kouwen, 2010) is a distributed model of intermediate complexity that only needs precipitation and temperature as forcing, as opposed to MESH and GEM-Hydro which need additional atmospheric variables (Table 1). WATFLOOD has been employed by Pietroniro et al. (2007) over the Great Lakes watershed.

In this project, WATFLOOD and MESH are implemented with a 10 arcmin (≈ 20 km) spatial resolution (both for their LSS and routing schemes), while GEM-Hydro is implemented with a 10 arcmin resolution for the LSS and 0.5 arcmin (≈ 1 km) for the routing. Sensitivity tests (Gaborit et al., 2016) revealed that 2 and 10 arcmin resolutions for SVS lead to quite similar performance in terms of streamflow at the outlet, while a substantial amount of computational time is saved when running the coarser resolution. WATROUTE produces outputs of similar quality when it is implemented at a low (10 arcmin for MESH and WATFLOOD) or high (0.5 arcmin with GEM-Hydro) resolution, as long as the catchment size is not too small compared to the horizontal resolution of the routing scheme, but the high-resolution version is preferred in GEM-Hydro for consistency with the WCPS-GLS (Durnford et al., in preparation) recently developed at ECCC. Hence, the higher resolution GEM-Hydro's routing scheme is not expected to give GEM-Hydro any advantage in comparison to the other models.

The internal time-step used for GEM-Hydro is 10 minutes, which slightly improves streamflow simulations in comparison to a 30 min. time-step (see Gaborit et al., 2016). Further reducing it does not improve the results. The internal time-steps used for MESH and WATFLOOD are respectively equal to 30 and 60 minutes. Table 1 summarizes the main specificities of the models and the required forcing data. Table 2 shows the datasets used for physiographic information.

As the GEM-Hydro suite (including WATROUTE) is quite demanding in terms of computational time, it was decided to test a configuration of GEM-Hydro in which WATROUTE is replaced by a Unit Hydrograph (UH) during calibration, and which is here forth referred to GEM-Hydro-UH. The UH allows the estimation of the streamflow at the basin outlet by partitioning the basin averages of runoff and recharge in time. The same WATROUTE LZS formulation is used in GEM-Hydro-UH in order to estimate stream recharge. The UH only requires a decay parameter corresponding to the lag or response time of the considered catchment, which controls the delay between the rainfall event and the resulting streamflow peak. It is estimated with the Epsey method (Almeida et al. 2014), which requires the catchment area, perimeter, and the maximum and minimum elevations along the catchment main river. The UH lag-time is also used as a free parameter during calibration (Table 3). It is inspired from the UH applied to the routing storage of GR4J (Perrin et al., 2003), but is employed here at an hourly time-step. This framework allows a considerable reduction of computational time dedicated to calibration.





Hydrographs resulting from GEM-Hydro and GEM-Hydro-UH can be very similar (Fig. 2). Finally, the SVS parameters identified by calibrating GEM-Hydro-UH are next transferred to the full version of GEM-Hydro, which then only needs WATROUTE Manning coefficients to be adjusted (if needed) in order to mimic the optimal hydrographs obtained with GEM-Hydro-UH.

The version of WATROUTE used in this work with GEM-Hydro relies on spatially-varying Manning values derived from physiographic information (i.e., land use), and on spatially-constant values (i.e. the same everywhere inside a given watershed) for the two LZS coefficients. These values were manually adjusted in order to be suitable to the whole GRIP-O area (Fig. 1), and hereafter referred to as the standard values for WATROUTE. In contrast, WATFLOOD relies on spatially-constant values for the Manning and LZS coefficients, which are adjusted during the automatic calibrations (see

Table 4). In SA-MESH, 5 river classes are defined based on spatial attributes, and each class possesses its own Manning coefficients which are adjusted during calibration (Table 5). SA-MESH does not include the LZS representation. This configuration difference between the distributed models is not envisioned to give GEM-Hydro any advantage, as comparisons were made between using fixed or spatially-varying Manning values with GEM-Hydro, leading to the conclusion that performances could be the same in both cases after a few manual adjustments (see Gaborit et al., 2016).

### 1.2 Study area and data

  The GRIP-O spatial framework is defined on Fig. 1. A more detailed description of the area is available in Gaborit et al. (in Press).

  The Lake Ontario basin (Fig. 1) covers 83 000 km$^2$, of which 19 000 km$^2$ is the lake surface. All upstream water arriving through the Niagara River is excluded to focus only on the lateral runoff component of Lake Ontario NBS (see

Introduction). The US/Canada border follows the Niagara River, the middle of Lake Ontario, and the St.-Lawrence River down to Cornwall regulation dam, the outlet of Lake Ontario. Apart from some major cities (e.g. Toronto), the catchment is mostly rural (agriculture, pasture, forest), as shown in Danz et al. (2007).

  Streamflow time series were selected based on their duration and proximity to the lake shoreline. Of the 30 selected sites (Fig. 1), 27 have no missing data, 2 are complete at 94%, and one at 80% over the GRIP-O period. Nearly 70% of the

25 total Lake Ontario watershed is gauged by the selected sites. Most of the rivers are regulated in some ways, mainly for hydropower and flood mitigation, but these structures, which cannot be represented in lumped models, did not prevent them from reaching good performances (Gaborit et al., in Press). Performances were not better for unregulated subbasins than for regulated ones, which is due to the generally simple degree of regulation involved (i.e., regulation generally consists of artificial lakes with a simple weir at their outlet). Therefore, no effort was made to represent in a detailed manner the

30 artificial structures of the region in WATROUTE. Moreover, the small diversions occurring to fill some canals in the region, or even the aquifers which can contribute significantly to baseflow (Singer et al., 2003; Kassenaar and Wexler, 2006), do not prevent lumped models from reaching good performances. This is helpful to this study, yet the flow values involved in the diversions would a priori still have to be taken into account when estimating Lake Ontario's NBS.



The physiographic data required by the distributed models under study consist of soil texture, land use / land cover, Digital Elevation Model (DEM), and flow direction grids. Table 2 lists the datasets used to provide the physiographic and atmospheric inputs required by the models. 26 land cover classes are defined in GEM-Hydro, while WATFLOOD and MESH rely only on 7 of them, which are aggregations of GEM-Hydro classes. Soil textures are from the Global Soil Dataset

for Earth system modeling (GSDE; Shangguan et al., 2014), which contains information down to 2.8 m. However, soil texture is calibrated for MESH (Table 5). Soil texture was not calibrated for GEM-Hydro-UH, but some hydraulic parameters, which are derived from soil texture, were calibrated (Table 3). WATFLOOD does not need soil texture information (Table 2). By default, the maximum soil depth is defined as 1.4 m in GEM-Hydro, 4.1 m in MESH, and not defined in WATFLOOD. The maximum soil depth is calibrated in GEM-Hydro and SA-MESH (Table 3 to Table 5).

Sensitivity tests performed with GEM-Hydro (Gaborit et al., 2016) indicated that its outputs have a limited sensitivity to the maximum soil depth value, given that it is higher than 1 m.

Precipitation forcing consists of 24-hourly accumulations from the Canadian Precipitation Analysis (CaPA version 2.4b8). Over the period of interest, CaPA consists of precipitation fields modeled by the Canadian Regional Deterministic Prediction System (RDPS, ≈15 km resolution), corrected by local rain gauge observations (Lespinas et al., 2015). CaPA

provides both 6-h and 24-h accumulations. The 24-hour accumulations were preferred to the 6-h CaPA data because fewer observations (about twice less) are used in the 6-h product to correct the model fields of precipitation, especially over the US part of the domain. The daily CaPA accumulations were disaggregated on an hourly time-step by following the temporal pattern of hourly precipitation from the RDPS (Carrera et al., 2010). The remaining atmospheric forcings (Table 1) are taken from RDPS outputs, using short-term forecasts having lead time of 6 to 18 h.

**1.3 Calibration strategy**

The GRIP-O experiment extends from June 1st, 2004 to September 26th, 2011. Calibrating a hydrologic model over a period of four to five years is generally deemed sufficient to achieve reasonable model robustness (e.g. Refsgaard et al., 1996). The calibration period thus ranges from June 1st, 2007 to September 26th, 2011 (4.5 years). Validation is from June 1st, 2005 to June 1st, 2007 (2 years), and spin-up from June 1st, 2004 to June 1st, 2005 (1 year). The objective function is

25 the Nash-Sutcliffe criterion (Nash and Sutcliffe, 1970) computed on the square-root of the observed and simulated time series, in order to avoid over-emphasizing peak-flow events - here forth referred to as "NSE √". These decisions are consistent with the lumped modelling decisions made for GRIP-O in Gaborit et al. (in Press). Other evaluation criteria used in this study consist in the common Nash-Sutcliffe criteria (NSE), the Nash criteria calculated over the log of the flows ("NSE Ln"), and a Percent Bias criteria (PBIAS, equation 1) assessing the simulation's overall water budget fit: a positive

value denotes a general tendency to underestimate flows, and vice-versa.

$$PBIAS = \frac{\sum_{i=1,n}(Qobs_i - Qsim_i)}{\sum_{i=1,n}(Qobs_i)} * 100 \qquad (1)$$





All metrics are evaluated at the daily time-step. Calibration relies for all models on the Dynamically Dimensioned Search (DDS) algorithm (Tolson and Shoemaker, 2007). Calibration cost did not allow models to be calibrated locally for all GRIP-O subbasins (Fig. 1). One local calibration takes between 2 and 5 days of computation. Table 3 to Table 5 list the free parameters of the models. Different paradigms were used to calibrate them. GEM-Hydro-UH was calibrated using

multiplicative coefficients that adjust the spatially-varying values of a given parameter, leading to a reasonable number of free parameters (16) while preserving spatial variability. MESH was implemented calibrating the 12 free parameters of its 5 different GRUs in an independent manner, thus resulting in 60 free parameters. WATFLOOD had the lowest number of free parameters during calibration, and involved calibrating parameter values which are valid for the entire subbasin (no spatial variability) or for one of the three main land cover types considered inside the model, i.e. bare ground, snow covered ground,

or other grounds (Table 4).

It is important to emphasize that the approach used to calibrate GEM-Hydro may result in unrealistic values for some parameters, as the multiplicative coefficients could bring them beyond the range of physical coherence. More precisely, soil water content thresholds and albedo (Table 3) cannot be higher than 1. Therefore, these values were constrained to realistic ranges after they were adjusted by the calibration algorithm.

The initial parameter values were either set to default ones that generally provide satisfactory results for the model (GEM-Hydro-UH, Table 3) or to random values (WATFLOOD, MESH). The number of maximum model runs allowed depends on the model being used. For example, 400 runs revealed sufficient for GEM-Hydro-UH (Sect. 2.2) in the sense that no significant performance improvement was achieved beyond. This is because the number of GEM-Hydro-UH free parameters is relatively low (16, Table 3). The DDS algorithm is very efficient in the sense that it adjusts the search behavior

to the maximum number of objective function evaluations (model runs) in order to converge to good quality solutions (Tolson and Shoemaker, 2007).

A maximum of 1000 model runs was used to calibrate MESH and of 1500 for WATFLOOD. Finally, the calibration strategy used for MESH consists of an improved and reliable strategy based on the work of Haghnegahdar et al. (2014). Despite the random initial values used for MESH and WATFLOOD, only one calibration trial was performed for

each of the models on a given subbasin. Even though the three models studied here were not calibrated using the same number of free parameters and the same maximum allowed model runs, it is assumed that the calibration strategies employed allow each model to come very close to its optimal performance for a given subbasin and the time period considered. Indeed, the strategy used for each of the three models is the result of expert knowledge and always involves parameters affecting the whole range of the main hydrological processes, i.e. evaporation, snowmelt, infiltration, soil transfer, and time to peak

(channel friction). It is thus logical to use different strategies for each of the models as these do not involve the same parameters, land use classification, or even physical processes. The most important methodological consistencies for achieving a fair comparison between models include, in our view, a common calibration algorithm and objective function, along with common physiographic and forcing data.



Finally, some subbasins in Fig. 1 have several gauge stations. In this case, the most-downstream observed flows on independent tributaries are summed and then extrapolated to the whole subbasin using the Area Ratio Method (ARM; Fry et al., 2014). The resulting "synthetic" flows were considered as observations for GEM-Hydro-UH calibration over the whole subbasin. This methodology was applied to all subbasins with more than one most-downstream gauge (identified with the "N/A" mention for the station attribute in Table 6) for consistency with the calibration experiments performed in the first GRIP-O study (see Gaborit et al., in Press). For these subbasins, the true gauged fraction is specified in Table 6.

## 1.4 Strategy for ungauged areas

The ultimate objective of the GRIP-O project consists on improving simulated Lake Ontario NBS, which calls for estimating runoff from all ungauged areas. For that sake, calibration was performed using GR4J and GEM-Hydro-UH models on the GRIP-O gauged area, and the resulting parameter sets were transferred to the same models but when implemented for the whole Lake Ontario watershed, including its ungauged parts (Fig. 1). The GRIP-O gauged area consists of the true gauged area (Fig. 1), plus the ungauged areas of the gauged subbasins including multiple gauge stations. This is because with local models (as with GEM-Hydro-UH) and in the case of subbasins with multiple gauges, the implementation was performed over the whole subbasin, including its ungauged part (see above). Therefore, the gauged area considered in this section and referred to as the "GRIP-O gauged area" actually covers 88.5% of the whole watershed.

The approach based on calibration for the GRIP-O gauged area and parameter transfer to the whole Lake Ontario watershed was preferred to other possible alternatives mainly for two reasons: it allows calibrating the models using close approximations of observed flows (the area used for calibration is gauged at 88.5%) instead of less reliable flow estimations for the whole watershed (gauged at 70%), and to take into account rainfall over the ungauged areas as well as rainfall over the gauged areas, or, in other words, to use the best approximation of rainfall. Yet this methodology involves two implementations of each model: one for the gauged part of the watershed and one for the whole area (Fig. 1).

It was demonstrated in the first GRIP-O paper (see Gaborit et al., in Press) that a unique (i.e., single) GR4J model calibrated over a large area could lead to runoff estimates of similar quality than with multiple models implemented over local subbasins. This was also demonstrated by Croley (1983) with the LBRM. A single (global) model has the advantage of requiring only one implementation and calibration, whereas local models require multiple implementations and possibly multiple calibrations. Therefore, a unique GR4J model was implemented twice, one over the GRIP-O gauged area (see above) for calibration, and one over the whole Lake Ontario watershed.

With GR4J, the parameter transfer protocol is straightforward as we end up with a unique parameter set for the unique model. However, GEM-Hydro-UH was here implemented in a local manner, i.e., for each of the gauged GRIP-O subbasins. When it is calibrated locally for each of the gauged subbasins, we end up with specific parameter sets for each of the subbasins, making the reliability of any parameter transfer very low (Sect. 2.1). Therefore, another strategy was chosen to calibrate GEM-Hydro-UH: global calibration.



Global calibration consists in finding a unique trade-off parameter set that allows to simultaneously improve performances for all subbasins (Ajami et al., 2004; Haghnegahdar et al., 2014; Gaborit et al., 2015), whereas local calibration consists in finding each subbasin's optimal parameter set. Local calibration logically leads to the optimal performances for a given subbasin, but global calibration may lead to temporal robustness (Gaborit et al., 2015) and spatial

consistency of the parameter values, because they are either fixed or adjusted the same way over the total area under study. Local calibration, on the other hand, because of equifinality and experiment imperfections (model processes, forcing data, observed flows, etc.), may compensate for simulation errors and lead to parameter sets that do not work well when transferred to other (even neighbor) subbasins, as tends to suggest the fact that very different parameter sets were obtained here with the local calibrations of GEM-Hydro-UH (Sect. 2.1).

For GR4J, local calibration was used but for a unique implementation on the complete GRIP-O gauged area. The objective function associated to global calibration of GEM-Hydro-UH is as follows:

$$OF = \sum_{i=1}^{N}(1 - \frac{Nloc_i}{Nglob_i}) \qquad (2)$$

with $Nloc_i$ the NSE $\sqrt{}$ value calculated from the local calibration on subbasin $i$, and $Nglob_i$ the NSE $\sqrt{}$ calculated from the global calibration on subbasin $i$. This objective function aims minimizing differences between performances obtained from

15 global and local parameter sets. However, as GEM-Hydro-UH was not locally calibrated for all of the 14 GRIP-O sub-basins, performances obtained with local GR4J calibrations (Gaborit et al., in Press) were used when needed (justifying the use of that model in this study).

Moreover, a supplemental free parameter was used for GEM-Hydro-UH during global calibration (in addition to those in Table 3), namely the percentage of completely impervious urban areas. This value was fixed to 0.33 during local

calibrations, implying that 33% of liquid precipitation or snowmelt over urban covers was automatically considered as runoff with no chance to infiltrate. This arbitrary value comes from a former study calibrating the SWMM 5 model over urban subbasins in Québec City, Canada (Gaborit et al., 2013). With local calibration, good performances could be reached, using this fixed value, even for "urban" subbasins (such as subbasins 14 and 15 in Sect. 2.1) as the effect of urban surfaces could be accounted for by other free parameters in Table 3. Moreover, this additional parameter helps to distinguish between

natural and urban surfaces for global calibration. The calibrated value of the urban cover fraction, which is completely impervious, is equal to 0.69 after global calibration (Table 7). This does make sense as the urban areas around the shore of Lake Ontario generally correspond to high-density areas, such as for the city of Toronto. Note also that with global calibration, the response time parameter controlling the UH duration (Table 3) was replaced with a multiplicative factor adjusting the default response times of all local subbasins.

Models were finally implemented over the whole Lake Ontario watershed (Fig. 1), and runoff simulations performed with the parameter sets calibrated over the GRIP-O gauged area. GEM-Hydro was selected for this task instead of GEM-Hydro-UH since it was more straightforward and a priori more realistic (see further) to use Watroute instead of the simple UH for the ungauged areas of the lake Ontario watershed. In GEM-Hydro, standard Manning coefficients were used



in Watroute, while the lag-time of GEM-Hydro-UH was adjusted during calibration. But it was assessed that simulations with GEM-Hydro (calibrated SVS and LZS parameters and standard Manning values) were very close, both in terms of hydrographs and performances at the gauged sites, to those from the calibrated GEM-Hydro-UH. Performances are generally even slightly better with GEM-Hydro (despite the standard Manning values) than with GEM-Hydro-UH (see Table 8).

## 2 Results and discussion

The comparison between GEM-Hydro and GEM-Hydro-UH is first presented to demonstrate the relevance of the UH approach to save the computation time associated with running the routing model of GEM-Hydro. Score improvements obtained by calibrating GEM-Hydro-UH for several subbasins of Lake Ontario watershed are then presented, followed by a performance comparison for all models. Finally, the methodology proposed with GEM-Hydro and the lumped GR4J model to simulate streamflows for the ungauged parts of the Lake Ontario watershed is evaluated.

Figure 2 presents the hydrographs simulated for the Moira river (subbasin 11 in Fig. 1), with SVS default parameters, standard Watroute parameter values in the case of GEM-Hydro, and a UH lag time estimated with the Epsey method in the case of GEM-Hydro-UH. As can be seen from this figure, GEM-Hydro-UH is able to produce streamflow simulations which are very close to those obtained with GEM-Hydro, underlying the relevance of such an approach to save computational time. Between the uncalibrated GEM-Hydro and GEM-Hydro-UH performances and over the different GRIP-O subbasins, the average absolute difference in Nash √ was 8% with the worst difference being 21%. See also Table 8 for a comparison between the calibrated GEM-Hydro and GEM-Hydro-UH models when looking at performances for the total GRIP-O gauged area. A complete GEM-Hydro run over the GRIP-O calibration period (4.5 years) takes about 48 hours, while the GEM-Hydro-UH version requires only 2 hours over the same period. despite the former relies on standard Manning coefficients.

### 2.1 GEM-Hydro-UH local calibrations

This section presents GEM-Hydro-UH performances (Fig. 3) either with its default parameter values or after its local calibration on Lake Ontario subbasins, which characteristics are given in Table 6.

As can be seen from Fig. 3, calibration provides substantial improvements in NSE √ values. Similar results were obtained for NSE and NSE Ln (although these results are not shown), and a lower improvement for PBIAS. Interestingly, all uncalibrated NSE √ are above zero (Fig. 3), and even satisfactory for subbasins 10 and 11. This is encouraging for ungauged subbasins applications.

Calibration also improves GEM-Hydro-UH Snow Water Equivalent (SWE) simulations but to a lesser degree than for the streamflow (not shown). Calibration does influence evapotranspiration, but no observations are available to evaluate this model output. For example, for the Moira River, the mean subbasin annual evapotranspiration (over the calibration period) is equal to 527 mm and to 647 mm, before and after calibration respectively. The robustness of the model is also deemed very good, since performances do not substantially deteriorate between calibration and validation (Table 8).



Calibrated parameter values are quite different from one subbasin to the other (even for neighbor subbasins), which may be due to equifinality (different parameter sets can lead to similar simulations) but also to the anthropogenic streamflow regulations. Table 7 presents the ranges of the final parameter values obtained with local calibration. This strongly limits the potential for parameter transferability to ungauged subbasins (Razavi and Coulibaly, 2012; Parajka et al., 2013). As explained in Sect. 1.4, global calibration can help overcoming this by leading to a spatially-coherent parameter set. Results of such an approach are presented in Sect. 2.3.

Calibrated GEM-Hydro-UH performance values are generally very close to those obtained with GR4J and CaPA precipitation (Fig. 3): the mean absolute difference in Nash √ values is 6.1%, with the maximum being 15%. This is very encouraging as the performance benchmark set by GR4J simulations is most of the time quite high and hard to attain for other models. Moreover, as new improvements are in progress for SVS (see below), it is probable that GEM-Hydro-UH and GEM-Hydro will even be able to surpass GR4J in terms of performance in the near future.

## 2.2 Inter-comparison of all models

This section aims at comparing MESH, WATFLOOD, and GEM-Hydro-UH performance values. The calibration strategy used for each of them is described in Sect. 1.3. Note that MESH was only calibrated on the Moira and Black Rivers, and WATFLOOD on the Moira, Black, and Salmon Rivers. Calibration and validation performances are presented in Fig. 4 and calibrated hydrographs, in Fig. 5.

It was deemed uninformative to present the calibrated parameter values since they are highly location dependant and subject to the equifinality issue (see previous section). Table 7 however highlights the final parameter ranges for GEM-Hydro-UH. Overall, GEM-Hydro-UH outperforms MESH and WATFLOOD, both in calibration and validation (Fig. 4). The robustness of the models is generally quite good, but less so for MESH on the Black River (subbasin 7 in Fig. 4).

When looking closely at the Moira River hydrographs (Fig. 5), important differences arise between the models. For instance, WATFLOOD has a more flashy behavior and tends to overestimate peak flow events, MESH generally underestimates flows, and GEM-Hydro-UH lays somewhere in between. Peak flow events associated to the spring freshet are generally better represented by MESH, which may be due to a better representation of the soil freezing and melting processes occurring in CLASS (MESH LSS).

It is possible that the differences in model performance may be explained by the different calibration strategies used for each model, and that better performances could be obtained with MESH and WATFLOOD for these watersheds, although the calibration details were in each case determined by an expert user of each model. The optimal calibration strategy, as well as the number of free parameters, could be revisited for each model in order to see if this explains the above differences, but this is quite beyond the scope of the paper.

Even if the intercomparison is obviously limited in the number of available test cases, it allows highlighting the mandatory need of calibrating hydrologic models, that models have unique behaviors that translate in substantial differences





in hydrographs, and that each of the models could benefit from some strengths of its competitors. For example, SVS would likely benefit from the implementation of the soil freezing and melting processes that are present in CLASS.

Results however strongly indicate that SVS can compete with more established Canadian models for simulating streamflow. In the coming years, after SVS becomes operationally implemented within ECCC's GEM-based NWP systems, it will be possible to obtain useful streamflow predictions by simply post-processing the runoff output from GEM using a unit hydrograph, or by routing these time series using a more sophisticated routing scheme.

## 2.3 Runoff estimation for the whole Lake Ontario watershed

The parameter values identified from the global calibration are presented in Table 7, along with the ranges resulting from local calibrations. See Sect. 1.4 for more information about methodology related to global calibration.

It can be seen from Table 7 that final global parameters generally lay inside the intervals obtained from local calibration, highlighting the trade-off found by global calibration.

GEM-Hydro-UH results are given first for each gauged subbasin, in order to compare global calibration, local calibration and default parameters (Fig. 7), followed by GR4J and GEM-Hydro results for the GRIP-O gauged area and the whole Lake Ontario watershed (Table 8 and Fig. 8).

GEM-Hydro-UH performances are lower with global calibration than with local calibration, as expected, and sometimes even lower after global calibration than with the default parameters for some subbasins (notably 10 and 11, Fig. 6). However, performances are satisfactory for most of the 14 GRIP-O subbasins with a single parameter set, which confirms that global calibration fulfilled expectations. Given that it takes between 2 to 5 days to achieve a local calibration, global calibration, which was completed in 10 days, allows to save a substantial amount of computational time. Furthermore and as previously stated, global calibration favors the spatial consistency of parameters and facilitates parameter transfer to ungauged areas, whereas there is no a priori best manner to transfer parameter values obtained from local calibration (Razavi and Coulibaly, 2012; Parajka et al. 2013). In this study, the strategy related to parameter transfer to the ungauged subbasins is based on spatial proximity, which was already identified as among the best parameter transfer methods for this type of climate in Canada (Razavi and Coulibaly, 2012).

Performance evaluation for the total GRIP-O gauged area (Table 8) shows that GR4J is better than GEM-Hydro-UH in calibration, but worse in validation. GEM-Hydro-UH leads to a very satisfactory performance, but most importantly to a better streamflow simulation than GR4J in terms of dynamics (see Fig. 7). Note that the smoother GR4J behavior is not due to the single model approach for the whole area, as a similar behavior occurred when aggregating simulations from local GR4J models (Gaborit et al., in Press). This smooth behavior seems inherent to the lumped attribute and concepts of GR4J. As depicted in Table 8, performances for the GRIP-O gauged area obtained with GEM-Hydro are close to those obtained with GEM-Hydro-UH, despite being lower for the former, which comes from the standard (uncalibrated) Manning coefficients used with GEM-Hydro, whereas the UH lag time was adjusted during the calibration of GEM-Hydro-UH.



Runoff simulations for the whole Lake Ontario watershed, including its ungauged areas, are very promising (Table 8). Even if runoff observations actually consist in this case in estimations based on the ARM, computed performances are a priori reliable given that the true gauged fraction of the total area is equal to about 70%. GEM-Hydro (and GEM-Hydro-UH) tends to overestimate streamflow total volumes (Table 8, PBIAS), while GR4J achieves a better estimation of the total runoff

volumes. The fact that GR4J is better than GEM-Hydro-UH in terms of PBIAS is attributed to the fact that GR4J consists in a single (global) model for the whole area considered. PBIAS values obtained with local GR4J models were poorer (Gaborit et al., in Press).

It is important to emphasize that for the whole watershed including its ungauged parts, runoff was estimated with GEM-Hydro instead of GEM-Hydro-UH, which means that streamflow simulations are available at all points inside the

domain, whereas GR4J only delivers estimations at the outlet. Moreover, even if the scores are slightly better for GR4J, the streamflow dynamics are generally better represented by GEM-Hydro, as is the case for example for the 2006 summer of Figure 7: GR4J represents a smooth streamflow recession, while GEM-Hydro-UH better follows the small peaks and drops occurring during the recession.

It is therefore argued that the methodology proposed here (global calibration of GEM-Hydro-UH and parameter

transfer to GEM-Hydro) is relevant, efficient, and reliable, provided that a large enough fraction of the total area is gauged. It could moreover be applied in different climatic contexts, regions, and with different models.

Simply extrapolating GEM-Hydro-UH simulated flows from the GRIP-O gauged area to the whole Lake Ontario watershed with the ARM leads to the exact same performances as those of the GRIP-O gauged area, because when doing so, we end up with both the simulated and observed flows being extrapolated the same way (i.e., with the ARM), which does not

change the scores at all. Based on these scores, it could be tempting to conclude that extrapolating the GEM-Hydro-UH flows to the whole Lake Ontario watershed leads to better results than transferring the calibrated parameters to GEM-Hydro over the whole Lake Ontario watershed, but it has to be reminded that for the whole watershed, observed flows are estimated with the ARM, which does not allow to completely trust the scores obtained (Table 8). No test was performed by implementing GEM-Hydro-UH over the whole basin.

Finally, Lake Ontario monthly NBS were estimated with the globally calibrated GEM-Hydro model, and results were compared both to the GLERL residual and component NBS estimates (Fig. 8). Residual NBS rely on the lake observed change in storage and streamflows for the Niagara and St-Lawrence rivers (DeMarchi et al., 2009). Component NBS used here are based on the GLERL Monthly Hydrometeorological Database (GLM-HMD; Hunter et al., 2015), which relies on observed data extrapolated with the ARM for runoff, on observed data interpolated with the Thiessen polygon method for

overlake precipitation, and on the Large Lake Thermodynamics lumped Model (LLTM) for overlake evaporation. Component NBS estimates are updated on a regular basis. Data used in this work were updated on August 2nd, 2016. It is still unknown which of these two NBS estimation methods (i.e., residual or component method) is the most accurate (DeMarchi et al., 2009).





It can be seen that the cumulated NBS estimates derived from the calibrated GEM-Hydro model (using global calibration) stand between the component and residual NBS estimates, but are closer to the latter ones. It is however difficult to draw any conclusion regarding the bias of these estimation methods given the uncertainty associated with NBS estimates (DeMarchi et al., 2009). When comparing the GLM-HMD component NBS method to the calibrated GEM-Hydro simulation

on a component-by-component basis, the main difference between the two occurs for overlake evaporation, with evaporation from the component method being significantly lower than GEM-Hydro evaporation (not shown). This mainly explains why the NBS estimates from the component method are higher than the other estimates in Figure 8. But again, it is not possible to accurately evaluate overlake evaporation estimates given the lack of observations for this variable. The uncalibrated GEM-Hydro model results in cumulative NBS estimations which are below all other NBS estimations, which tends to suggest that

they are underestimated.

**Conclusion**

Our results indicate that the SVS LSS, as embedded in GEM-Hydro and GEM-Hydro-UH, provides a reasonable simulation of runoff to Lake Ontario. This result is encouraging because SVS is expected to replace ISBA in ECCC operational models in the near future. However, there is still room to further improve SVS. For example, as illustrated while

comparing GEM-Hydro-UH, WATFLOOD and MESH, SVS may benefit from the implementation of soil freeze-thaw processes, the current absence of which is assumed to be partly responsible for SVS missing some of the runoff peaks in spring. A new snow module (ISBA-ES) is also being implemented into SVS, which currently relies on a simple force-restore approach.

According to the intercomparison experiment conducted on three subbasins, GEM-Hydro-UH and GEM-Hydro are

competitive to MESH and WATFLOOD. However, as a limited number of subbasins were used for the inter-comparison due to computational time limitations, no general model ranking can be derived from this study. Calibration has of course proven that it is mandatory to optimize model performances. The calibrated GEM-Hydro-UH performances are close to GR4J ones (Gaborit et al., in Press).

The potential benefits of global calibration have been demonstrated here. It achieves satisfactory performances for a

large area with a unique calibration and favors temporal robustness, spatial consistency, and parameter transferability. Global calibration of SVS is envisioned in future versions of ECCC's WCPS and has already proven interesting for different modeling platforms too, such as Hydrotel (Gaborit et al., 2015).

It is also envisioned to assess the benefits of SVS global calibration in improving weather forecasts, as a calibrated SVS could be coupled to ECCC's RDPS atmospheric model, and because a calibrated SVS version should improve surface

fluxes representation. Calibrating a LSS based on streamflow and then using it in an atmospheric model to improve weather forecasts has not been reported in the literature so far, to our knowledge.

Finally, an efficient and transferable methodology has been proposed to estimate runoff for ungauged parts of a watershed. However, the method is not applicable if the area is completely ungauged. For this, however, GEM-Hydro has





proven able to produce decent, generally satisfactory runoff simulations with default parameter values, except for areas with a high urban cover fraction, which needs further investigation.

In order to calibrate the GEM-Hydro model, its routing part was replaced by a simple UH during calibration, which saves a tremendous amount of computational time. The routing part of GEM-Hydro can be run afterwards, potentially adjusting the standard Manning values if needed. Lumped models have limited applications, while distributed ones can be useful to a number of environmental studies. Many distributed models do exist worldwide, each one possessing its own advantages and drawbacks, but also its own optimal implementation and calibration methodology, which makes a perfectly fair inter-comparison quite challenging, if not unrealistic.

This work successfully led to the implementation of an efficient distributed hydrological modeling platform for the land portion of Lake Ontario watershed, which has therefore become a readily testing ground for distributed models, for example for upcoming SVS improvements which are currently being implemented at ECCC and whose resulting benefits on streamflow simulations are dedicated to future work.

**Ackowledgements**

This work was supported by the IJC International Watersheds Initiative. The authors wish to thank Dorothy Durnford, Nasim Alavi, and Maria Abrahamovicz, from Environment and Climate Change Canada (ECCC), who provided support with the GEM-Hydro platform implementation, and Djamel Bouhemhem (ECCC), for informatics support. This is NOAA-GLERL contribution No. XXXX. The works published in this journal are distributed under the Creative Commons Attribution 3.0 License. This licence does not affect the Crown copyright work, which is re-usable under the Open Government Licence (OGL). The Creative Commons Attribution 3.0 License and the OGL are interoperable and do not conflict with, reduce or limit each other.

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



**Figures**

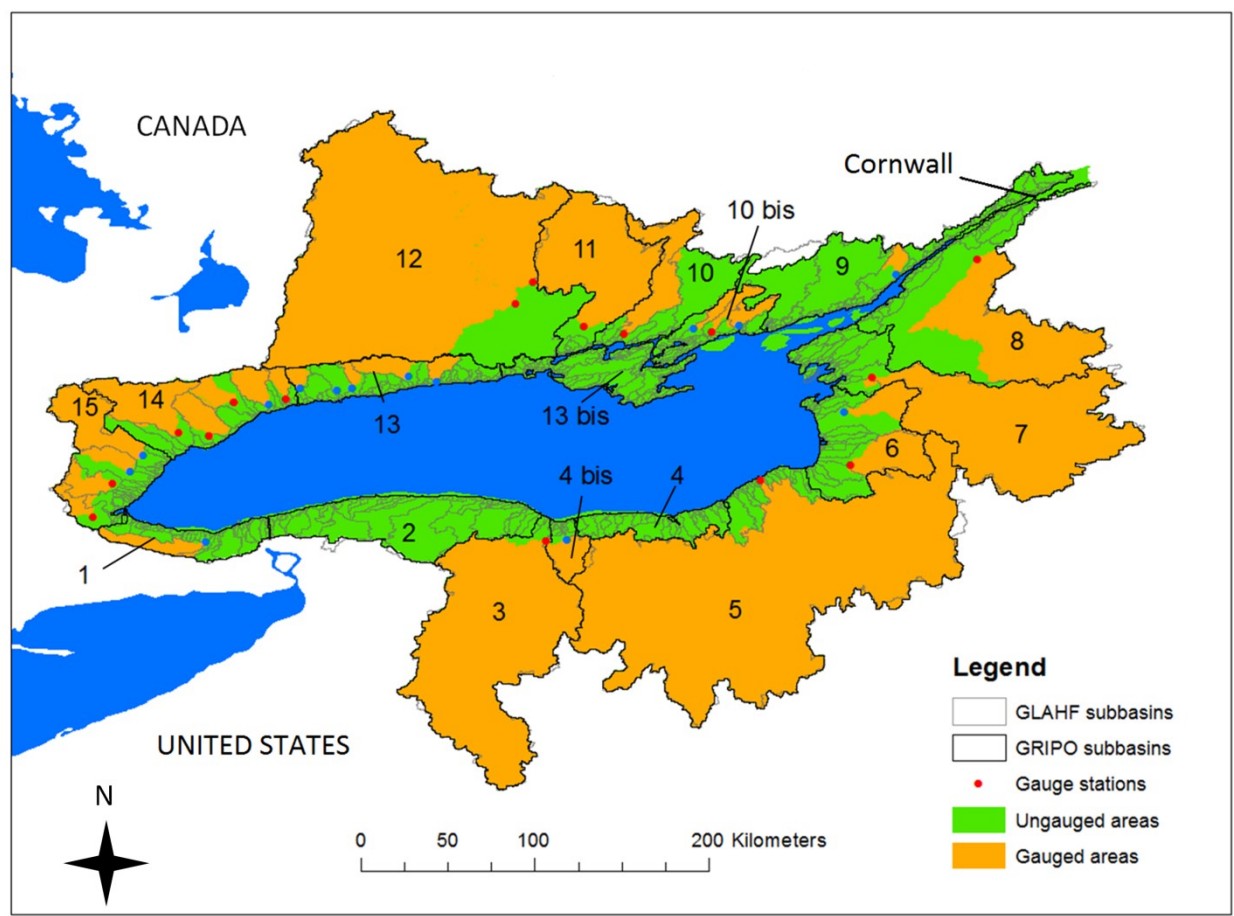

**Figure 1: GRIP-O spatial framework: Lake Ontario subwatershed delineation (GRIP-O subbasins). GLAHF subbasins are from the Great Lakes Aquatic Habitat Framework (Wang et al., 2015). Dots (blue for natural flow regimes and red for regulated regimes) are the most-downstream flow gauges selected for model calibrations.**



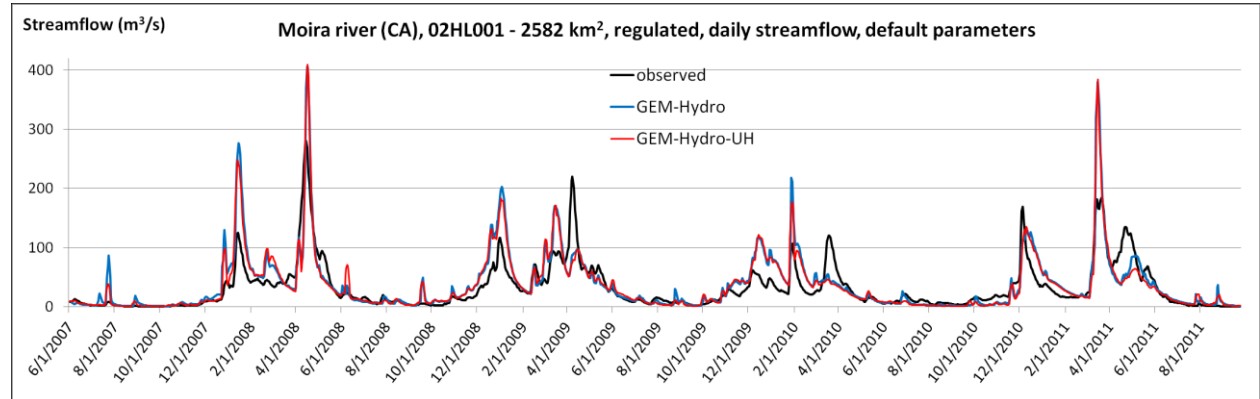

**Figure 2: Hydrographs from uncalibrated GEM-Hydro and GEM-Hydro-UH (Moira River).**

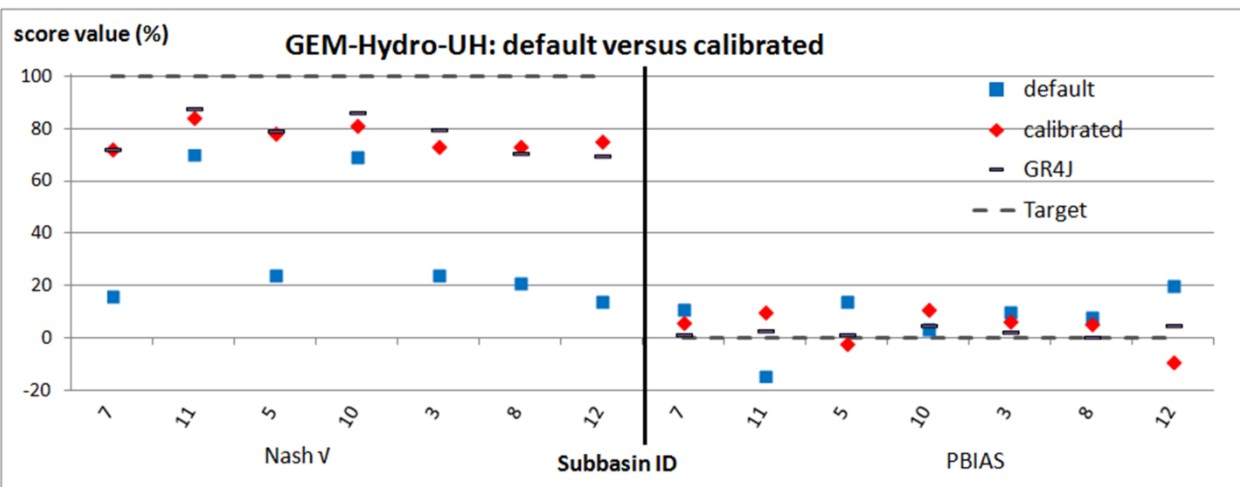

5    **Figure 3: Uncalibrated and calibrated GEM-Hydro-UH performances over the calibration period. Results are presented as NSE √ (left) and PBIAS (right), for many GRIP-O subbasins. The grey dashed line shows perfect scores and GR4J reference is displayed with black markers.**





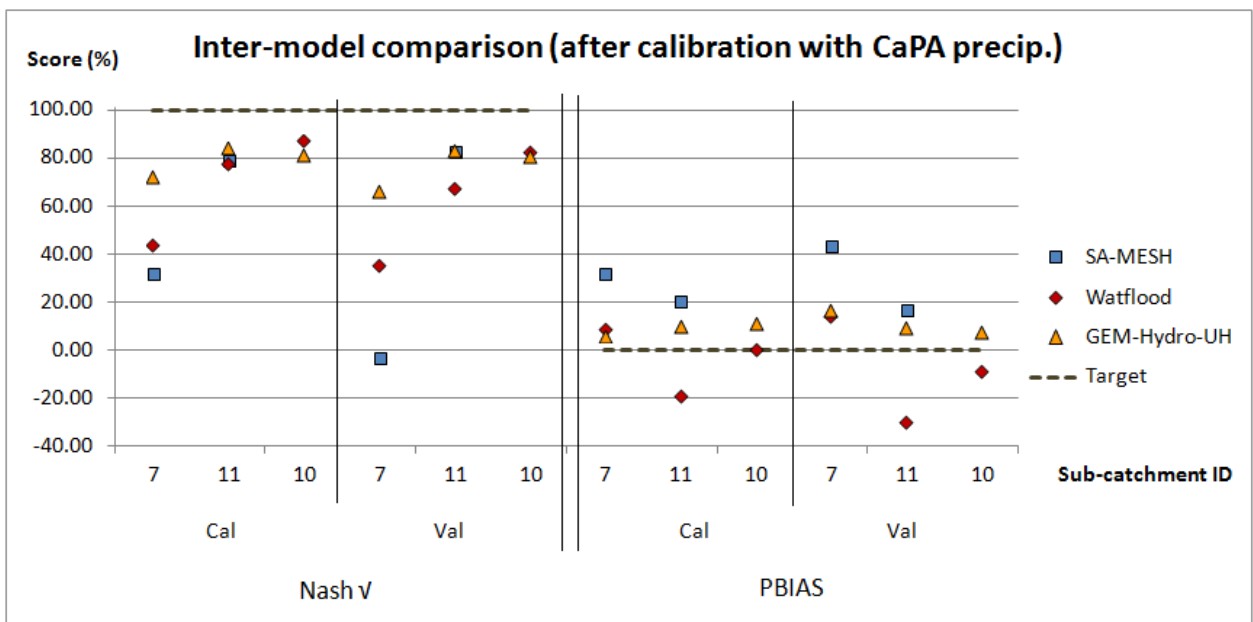

**Figure 4: Intercomparison for three GRIP-O subbasins (Table 6). MESH was not implemented on subbasin 10. Cal, Val: calibration and validation periods, respectively. Scores that would be achieved if models provided a perfect fit to observations are indicated by the dashed line and labelled "Target".**

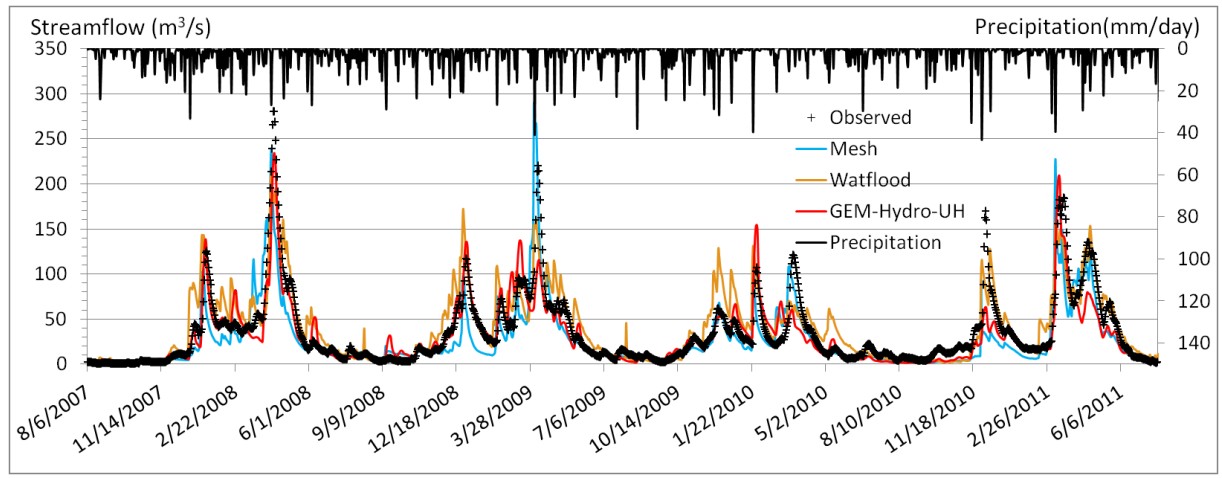

**Figure 5: Intercomparison for the Moira River (calibration period, CaPA pecipitation).**



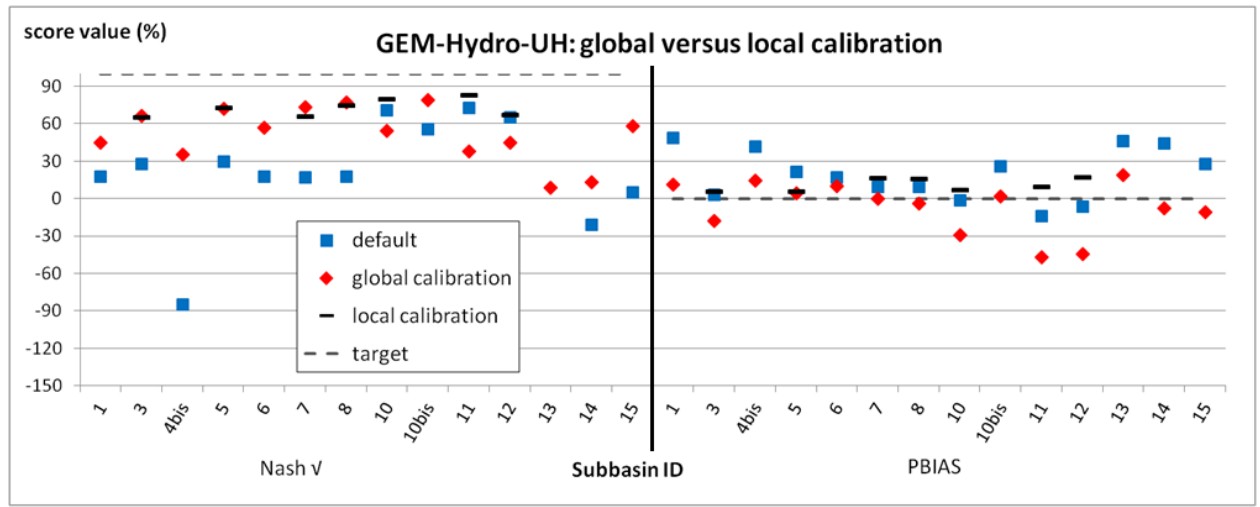

**Figure 6: GEM-Hydro-UH performances in validation for the 14 GRIP-O gauged subbasins (Fig. 1) with default, locally, and globally-calibrated parameter values. Perfect scores are shown.**

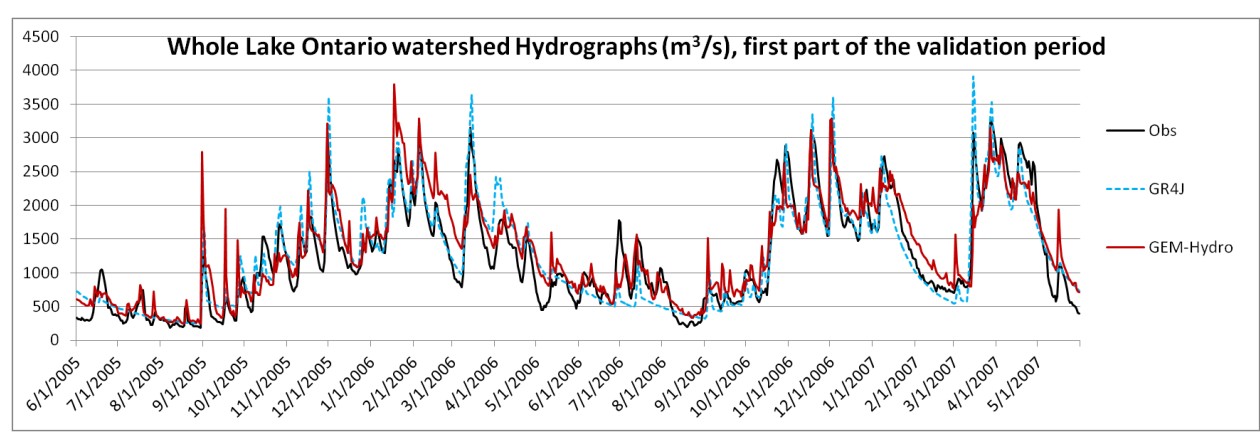

**Figure 7: Lake Ontario watershed runoff (including its ungauged areas, Fig. 1) for the validation period, comparing GR4J and GEM-Hydro.**



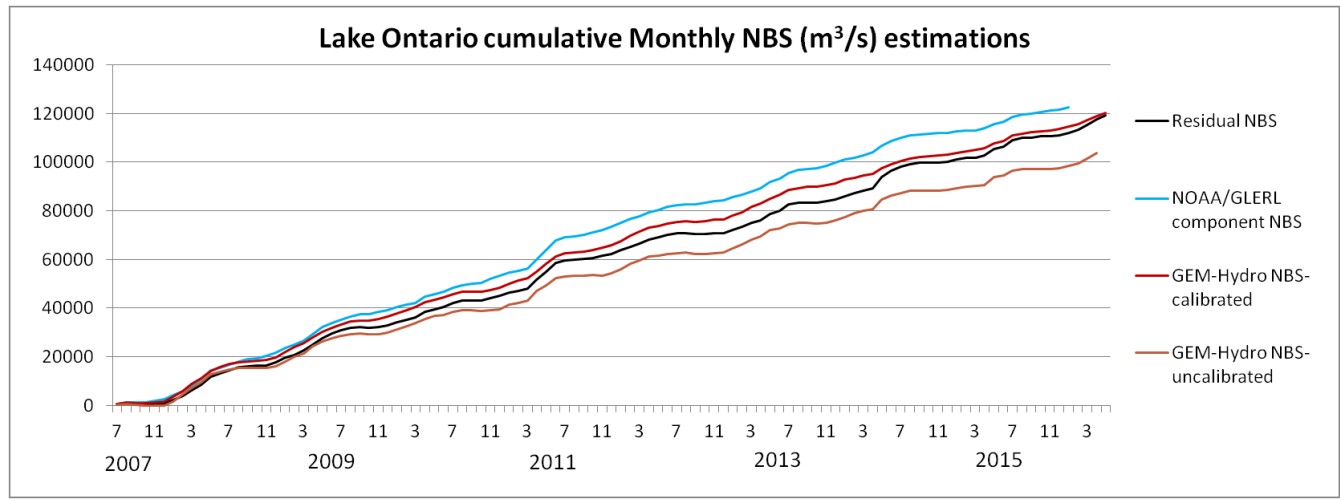

**Figure 8: cumulative Lake Ontario NBS estimates. See text for further details.**

**Tables**

**Table 1: Data requirements and model specificities. P: precipitation, T: temperature, H: humidity, R:, radiative forcings, W: wind,**

5   **Ps: pressure; LULC: Land Use / Land Cover, Topo: elevation data, Flow Dir: flow directions. Brackets indicate time-step used in this study.**

| Model name | Underlying theory | Spatial distribution | Time-step [min] | Forcing data | Physiographic data |
|---|---|---|---|---|---|
| WATFLOOD | Physical/Conceptual | Semi-distributed | Flexible [60] | P, T | LULC, Topo, Flow Dir |
| GEM-Hydro | Physical | Semi-distributed | Flexible [10] | P, T, H, R, W, Ps | LULC , Soil, Topo, Flow Dir |
| MESH | Physical | Semi-distributed | Flexible [30] | P, T, H, R, W, Ps | LULC, Soil, Topo, Flow Dir |



**Table 2: Data sources; NA: North America**

| Dataset/origin | Type of data | Coverage | Resolution/scale | Source |
|---|---|---|---|---|
| GSDE | soil texture | Global | ~ 1km (30") | Shangguan *et al.* 2014 |
| GLOBCOVER 2009 | land cover | Global | 300m (10") | ESA 2009 |
| HydroSheds | Flow directions | Global | ~ 1km (30") | USGS and WWF 2006 |
| SRTM | DEM | Global | 90m (3") | NGA and NASA 2000 |
| HyDAT | Gauge stations | CA | N/A | ECCC |
| NWIS | Gauge stations | US | N/A | USGS |
| CaPA v2.4b8 | Precipitation | NA | ~ 15 km | ECCC |
| RDPS | Atmospheric forcings | NA | 15/10 km | ECCC |





**Table 3: Information on GEM-Hydro-UH 16 free parameters; LZS: Lower Zone Storage; coeff. : coefficient; mult. : multiplicative; precip. : precipitation; param.: parameter; min.: minimum; max.: maximum.**

| Param. \ range | description | initial | Min. | Max. | Param. \ range | description | initial | Min. | Max. |
|---|---|---|---|---|---|---|---|---|---|
| HU_decay | response time (h) | 60.0 | 20.0 | 400.0 | LAI | Leaf-Area Index mult. coeff. | 1.0 | 0.2 | 5.0 |
| FLZCOEFF | LZS mult. coeff. | 1.0E-05 | 1.0E-07 | 1.0E-04 | Z0M | roughness length mult. coeff. | 1.0 | 0.2 | 5.0 |
| PWR | LZS exponent coeff. | 2.8 | 1.0 | 5.0 | TBOU | boundary between liquid and solid precip. (˚C.) | 0.0 | -1.0 | 1.5 |
| MLT | coeff. To divide snowmelt amount | 1.0 | 0.5 | 2.0 | EVMO | evaporation resistance mult. coeff. | 1.0 | 0.1 | 10.0 |
| GRKM | Horizontal conductivity mult. coeff. | 1.0 | 0.1 | 30.0 | KVMO | vertical conductivity mult. coeff. | 1.0 | 0.1 | 30.0 |
| SOLD | soil depth (m) | 1.4 | 0.9 | 6.0 | PSMO | soil water suction mult. coeff. | 1.0 | 0.1 | 10.0 |
| ALB | albedo mult. coeff. | 1.0 | 0.2 | 5.0 | BMOD | slope of retention curve mult. coeff. | 1.0 | 0.1 | 10.0 |
| RTD | root depth mult. Coeff. | 1.0 | 0.2 | 5.0 | WMOD | threshold soil moisture contents mult. coeff. | 1.0 | 0.1 | 10.0 |





**Table 4: Information on WATFLOOD 14 free parameters; LZS: Lower Zone Storage; coeff. : coefficient; mult. : multiplicative.**

| parameter | minimum | maximum | parameter | minimum | maximum |
|---|---|---|---|---|---|
| channel Manning's N | 0.01 | 1.0 | upper zone retention (mm) | 1.0 | 300.0 |
| LZS mult. coeff. | 1.0E-09 | 1.0E-05 | infiltration coefficient bare ground | 0.8 | 0.99 |
| LZS exponent coeff. | 2.0 | 3.0 | infiltration coefficient snow covered ground | 0.8 | 0.99 |
| melt factor (mm/dC/hour) | 0.1 | 3.0 | overland flow roughness coefficient bare ground | 1.0 | 75.0 |
| interflow coefficient | 1.0 | 100.0 | overland flow roughness coefficient snow covered ground | 1.0 | 75.0 |
| interflow coefficient bare ground | 1.0 | 200.0 | Interception evaporation factor | 0.1 | 75.0 |
| interflow coefficient snow covered ground | 1.0 | 200.0 | base temperature (dC) | -3.0 | 3.0 |





**Table 5: Information on MESH 60 free parameters: independent values are sought for each of the 5 model Grouped Response Units (GRUs; source: Haghnegahdar, 2015).**

| parameter | description | vegetation or river class (5) | minimum | maximum |
|---|---|---|---|---|
| ROOT | Annual maximum rooting depth of vegetation category [m] | crop and grass | 0.2 | 1.0 |
| | | Forest | 1.0 | 3.5 |
| RSMN | Minimum stomatal resistance of vegetation category [s.m$^{-1}$] | Crop | 60.0 | 110.0 |
| | | Grass | 75.0 | 125.0 |
| | | Forest | 100.0 | 150.0 |
| VPDA | Vapour pressure deficit coefficient | All | 0.5 | 1.0 |
| SDEP | Soil permeable (Bedrock) depth [m] | All | 0.35 | 4.1 |
| DDEN | Drainage density [km/km$^2$] | All | 2.0 | 100.0 |
| SAND | Percent sand content [%] | All | 0.0 | 100.0 |
| CLAY | Percent clay content [%] | All | 0.0 | 100.0 |
| RATIO | The ratio of horizontal to vertical saturated hydraulic conductivity | All | 2.0 | 100.0 |
| ZSNL | Limiting snow depth below which coverage is less than 100% [m] | All | 0.05 | 1.0 |
| ZPLS | maximum water ponding depth for snow-covered areas [m] | All | 0.02 | 0.15 |
| ZPLG | maximum water ponding depth for snow-free areas [m] | All | 0.02 | 0.15 |
| WFR2 | Channel roughness factor | All | 0.02 | 2.0 |





**Table 6: GRIP-O subbasins characteristics.**

| country | Subbasin # | Station | %_gauged | Area(km$^2$) | Flow regime | mean elev. (m) |
|---------|-----------|---------|----------|--------------|-------------|----------------|
| CA | 1 | 20_mile | N/A | 307 | natural | 198 |
| USA | 3 | Genessee | N/A | 6317 | regulated | 418 |
| USA | 4bis | Irondequoit | N/A | 326 | natural | 172 |
| USA | 5 | Oswego | N/A | 13287 | regulated | 259 |
| USA | 6 | N/A | 40 | 2406 | mixed | 264 |
| USA | 7 | Black river | N/A | 4847 | regulated | 471 |
| USA | 8 | Oswegatchie | N/A | 2543 | regulated | 250 |
| CA | 10 | Salmon_CA | N/A | 912 | regulated | 196 |
| CA | 10bis | N/A | 44.2 | 944 | mixed | 115 |
| CA | 11 | Moira | N/A | 2582 | regulated | 228 |
| CA | 12 | N/A | 88 | 12515.5 | regulated | 282 |
| CA | 13 | N/A | 40.3 | 1537.5 | natural | 178 |
| CA | 14 | N/A | 61.3 | 2689.4 | mixed | 209 |
| CA | 15 | N/A | 63 | 2245.8 | mixed | 263 |





**Table 7: Final parameter values or ranges after calibration; for global calibration, HU_decay consists of a multiplicative coefficient. See Table 3 for parameter definition.**

|  |  | HU_decay | FLZCOEFF | PWR | MLT | GRKM | SOLD | ALB | RTD |
|---|---|---|---|---|---|---|---|---|---|
| global calibration |  | 0.5 (mult) | 7.1E-07 | 2.3 | 0.7 | 6.7 | 0.9 | 1.0 | 3.7 |
| local calibration range | min | 46.0 | 1.4E-07 | 1.1 | 0.4 | 1.5 | 0.9 | 0.4 | 1.1 |
|  | max | 142.7 | 8.5E-05 | 4.2 | 1.5 | 13.1 | 4.6 | 2.0 | 3.9 |
|  |  | LAI | Z0M | TBOU | EVMO | KVMO | PSMO | BMOD | WMOD |
| global calibration |  | 1.9 | 3.9 | 0.4 | 1.8 | 2.9 | 1.5 | 0.7 | 1.4 |
| local calibration range | min | 0.6 | 0.2 | -0.9 | 0.6 | 1.0 | 1.2 | 0.6 | 0.6 |
|  | max | 4.6 | 3.8 | 0.5 | 3.5 | 9.4 | 9.4 | 1.5 | 2.8 |



**Table 8: performances for the GRIP-O gauged area and the whole Lake Ontario watershed (Fig. 1) with GR4J and globally-calibrated GEM-Hydro-UH and GEM-Hydro models. Cal., val.: calibration and validation periods, respectively.**

| | GRIPO gauged area: 53459.2 km$^2$ | | | | | | Lake Ontario basin: 68214.8 km$^2$ | | | |
| | GR4J | | GEM-Hydro-UH | | GEM-Hydro | | GR4J | | GEM-Hydro | |
| Scores (%) | cal | val | cal | val | cal | val | cal | val | cal | val |
|---|---|---|---|---|---|---|---|---|---|---|
| Nash | 82.4 | 84.6 | 80.1 | 83.4 | 79.8 | 80.5 | 82.9 | 85.5 | 81.8 | 82.0 |
| Nash √ | 84.7 | 85.5 | 83.0 | 86.6 | 78.5 | 82.4 | 84.4 | 85.0 | 80.5 | 83.7 |
| Nash Ln | 83.3 | 84.0 | 82.1 | 87.2 | 74.4 | 82.3 | 82.4 | 82.8 | 76.8 | 83.7 |
| Pbias | -0.3 | 1.5 | -9.0 | -8.1 | -13.1 | -10.9 | -2.2 | -1.2 | -10.3 | -8.2 |

