# Peer review of "A Hydrological Prediction System Based on the SVS Land-Surface Scheme: Implementation and Evaluation of the GEM-Hydro platform on the watershed of Lake Ontario"

_Hydrology and Earth System Sciences, 2016_

## Referee Comment (RC1) · Anonymous Referee #1 · 2 Dec 2016

General comments

The manuscript presents interesting topic, but in its current form it is very difficult to read and reads more like a technical report rather than a scientific paper. I have several comments, which might be considered for a revision:

1) The formulation of research hypothesis and novel scientific contribution is not clear. The first objective (as formulated in the manuscript) is to propose a methodology for

calibrating distributed hydrologic model, but the results refer to an apriori selected number of model runs (in model calibration), without testing whether this approach is better (and in which aspects) than some methodologies used for model calibration. The introduction refers to numerous previous studies but does not clearly indicate what unanswered question is investigated here. Just implementation reads more as a technical than a scientific question. In its current form it is formulated as a case study. Why it should be interesting for international audience of the journal? What can be learned/generalised from the results which will be interesting/relevant also for other regions in the world?

2) The number of model runs does not seem to be very large (i.e. adequate), so some more deep analysis and justification of such setup is needed. E.g. I wonder whether 400 runs/combinations for 16 model parameters are enough representative.

3) The 2. objective of the manuscript is to compare different models, but it is not clear why not ISBA and SVS are compared, as in the Introduction it is referred to the replacement of ISBA by SVS. This would allow to better demonstrate the potential of the new model/implementation.

4) The manuscript refers to many different other studies within the study region, but not all are relevant to the main objectives, so it distracts the reader from the main story line. Moreover, it is than not quite clear, in which respects is this study novel, so a more clear discussion of the novelty would be helpful. There are numerous references to studies in press or preparation, which does not allow to justify to what extent the study overlaps with previous/recent studies. I would suggest to consider streamlining the text flow, and do not refer much to studies which are not directly linked/relevant with the research questions studied here. For example references to lumped modelling results in the Study area section are not placed/relevant (well) here. Please consider also not to use so many abbreviations, because some parts are then very difficult to understand (e.g. p3., l14).

Summing up, the topic is interesting and within the scope of the journal. The manuscript however needs some revision and transformation to a more scientific than technical report.

Specific comments

1) Abstract: Please consider to be more specific (i.e. provide numbers, efficiencies, etc), particularly when referring to results found. The context part for the research does not have to be such long.

2) Introduction (p.4., l.5-10): It will be important to clearly formulate in which respect is this study new in comparison to the first GRIP-O study. Please consider also discuss/show how specific was the model performance and how similar/different it is with respect to this study.

3) P.5, l2.: Please consider be more specific about the calibration strategy of Haghnegahdar et al. (2014).

4) p.5, l.14-18: This part is messy and not clear. Please consider to revise.

5) P.5, l19-21: Why are the used time-steps different? Has it some implications for interpreting results?

6) P.6, l.27: Why is lumped model mentioned here? Are the findings (good performance) for the right reasons? The reference of Gaborit is not accessible so it is difficult to see.

7) P.7, l.6, l.9: Which hydraulic parameters? Is the maximum soil depth calibrated for each grid cell or entire domain?

8) P.7, l.19: what is RDPS?

9) P8, l.3: how many runs has typically local calibration?

10) P.8, l.14: Please be more specific how were the values contrained?

11) P.9, l.1-6: This part is not clear. On how many points is then the model verified/compared /calibrated?

12) Strategy for ungauged basins: Typically, the prediction in ungauged basins is verified by leave-one-out approach. How do the results compare with such method? Please consider to discuss.

13) P.11, l.28-29: Please consider to show some results supporting this statement.

14) P.16, l.17: Please consider to update XXXX.

15) P.21, l.5: "the most-downstream flow gauges" is not clear.

16) Table 1: Please be more specific what is radiative forcings

---

## Short Comment (SC1) · 21 Dec 2016

Please see the attached file (supplement file) for a version of the manuscript tracking the changes listed in the answers to reviewer (AR):

General comments

The manuscript presents interesting topic, but in its current form it is very difficult to read and reads more like a technical report rather than a scientific paper. I have several

comments, which might be considered for a revision:

1) The formulation of research hypothesis and novel scientific contribution is not clear. The first objective (as formulated in the manuscript) is to propose a methodology for calibrating distributed hydrologic model, but the results refer to an a priori selected number of model runs (in model calibration), without testing whether this approach is better (and in which aspects) than some methodologies used for model calibration. The introduction refers to numerous previous studies but does not clearly indicate what unanswered question is investigated here. Just implementation reads more as a technical than a scientific question. In its current form it is formulated as a case study. Why it should be interesting for international audience of the journal? What can be learned/generalised from the results which will be interesting/relevant also for other regions in the world?

AR: the introduction (and abstract) was modified to make the novelty of this contribution clearer. Therefore, the reader should now better understand that the aim of the study is not to find the best way to calibrate distributed models (which can always be improved and has almost an infinity of possibilities, making it quite impossible to find anyway), but the proposition of an efficient and reliable way to implement a certain type of distributed models over a large area with ungauged parts. This could be applied anywhere on Earth. This is a novel approach because global calibration is rarely employed in the Litterature, and because we propose a new way of calibrating Land Surface Schemes, i.e., by getting rid of the time-consuming routing part of distributed hydrologic models. Another research objective is to contribute to the GRIP framework, by comparing different models with the same forcings and implementation methodology in the Great Lakes region. In regard of the long introduction, we actually believe that it gives a good overview of hydrological research in the Great Lakes region, which may be of interest to a relatively large audience. However, an effort of synthesis was still put into the introduction and it was slightly shortened. The new introduction is pasted below:

[revised manuscript text omitted]

2) The number of model runs does not seem to be very large (i.e. adequate), so some more deep analysis and justification of such setup is needed. E.g. I wonder whether 400 runs/combinations for 16 model parameters are enough representative.

AR: this sentence was added close to page 8, line 20, to further justify the methodology used here: " The similarity of the performances obtained with GR4J and GEM-Hydro-UH (Fig. 3) supports the choice of the methodology used here, as GR4J was implemented with a maximum of 2000 model runs, three distinct calibration trials, and had an even lower number of free parameters (6, see Gaborit et al., in Press). "

3) The 2. objective of the manuscript is to compare different models, but it is not clear why not ISBA and SVS are compared, as in the Introduction it is referred to the replacement of ISBA by SVS. This would allow to better demonstrate the potential of the new model/implementation.

AR: As stated in the manuscript close to page 5, line 4, the comparison between ISBA and SVS has been performed and results are described in the technical note mentioned in the manuscript. Results are not shown here for the sake of brevity and because the comparison of ISBA and SVS is, to us, of a lesser importance to the community than the comparison between different models (MESH, WATFLOOD and GEM-Hydro-UH).

4) The manuscript refers to many different other studies within the study region, but not all are relevant to the main objectives, so it distracts the reader from the main story line. Moreover, it is than not quite clear, in which respects is this study novel, so a more clear discussion of the novelty would be helpful. There are numerous references to studies in press or preparation, which does not allow to justify to what extent the study overlaps with previous/recent studies. I would suggest to consider streamlining the text flow, and do not refer much to studies which are not directly linked/relevant with the research questions studied here. For example references to lumped modelling results in the Study area section are not placed/relevant (well) here. Please consider also not to use so many abbreviations, because some parts are then very difficult to understand (e.g. p3., l14).

AR: most of this comment has been treated in the answer to the first reviewer's comment. The reference to the first GRIP-O study will be accessible as the corresponding paper will be published very soon. References to lumped modeling are important to support the methodology used here to calibrate GEM-Hydro-UH with a small number of model runs, given that it leads to similar performances as with the lumped GR4J model which was implemented with 2000 model runs and even fewer free parameters. Moreover, it has to be emphasized that this study is also the second part of the GRIP-O

study. GR4J is also used in the objective function used for global calibration, as missing GEM-Hydro performances obtained with local calibration were replaced with GR4J ones (see page 10, line 16). Some abbreviations were removed from the manuscript.

Specific comments

[revised manuscript text omitted]

3) P.5, l2.: Please consider be more specific about the calibration strategy of Hagh-negahdar et al. (2014). AR: the paragraph was changed into the following, as the calibration strategy is explained in more details in the section "calibration strategy". The version of MESH used in this study relies on version 3.6 of the Canadian LAnd Surface Scheme (CLASS). Each grid cell is subdivided in a number of tiles, and each tile is classified as belonging to one of the five grouped response units (GRUs), based on its land-use/soil type combination. In this paper, we follow the local calibration strat-egy advocated by Haghnegahdar et al. (2014) for MESH (see section on calibration strategy).

4) p.5, l.14-18: This part is messy and not clear. Please consider to revise. AR: this paragraph was modified into this: The same was shown for WATROUTE which pro-duces outputs of similar quality be it implemented at a low (10 arcmin for MESH and WATFLOOD) or high (0.5 arcmin with GEM-Hydro) resolution, as long as results are evaluated for large enough catchments (i.e., catchments which spread over at least a few grid cells). However, the high-resolution WATROUTE version is preferred in GEM-Hydro for consistency with the WCPS-GLS (Durnford et al., in preparation) recently developed at ECCC. Hence, the higher resolution GEM-Hydro's routing scheme is not expected to give GEM-Hydro any advantage in comparison to SA-MESH and WAT-FLOOD.

5) P.5, l19-21: Why are the used time-steps different? Has it some implications for interpreting results? AR: Generally, the internal time-step is adjusted in order to be maximized while preserving numerical stability, which hence depends on the equations and numerical schemes (implicit or explicit) of a given model. Of course, the internal time-step however has to remain lower than the output frequency. This is why WATFLOOD uses a time-step of 60 min. (results are being output on an hourly basis). This does not rise any issue in regard of results' interpretation, because we here assess the performances on a daily basis, which is a temporal resolution well resolved by all considered models. In order to make it more clear to the reader, the following sentence was added: " The internal time-step of a model is generally maximized up to the desired output interval, provided that it satisfies numerical stability. In the GEM-Hydro version used in this study, a 10-min. time-step was required to achieve numerical stability, but a newer version now allows to increase it."

6) P.6, l.27: Why is lumped model mentioned here? Are the findings (good performance) for the right reasons? The reference of Gaborit is not accessible so it is difficult to see. AR: Yes, results indicate that the good performances of the models are obtained for the right reasons, or in other words that simulations closely follow the observed streamflow dynamics with realistic parameter values. Moreover, as explained in the first GRIP-O study, the parameters obtained by GR4J on regulated catchments (which contain reservoirs) reveal that the model increased the routing storage capacity, which makes sense. These results obtained with lumped models during the first GRIP-O study are mentioned because they allow to comfort the fact that regulation in the area of the Lake Ontario watershed generally involves simple management operations, and are close to the behavior of natural lakes. The reference will soon be published in the Journal of Great Lakes Research (final corrected proof has been sent) and the final version of this paper will include a reference which is possible to access. To make it more easy to read and more logical, the paragraph was reformulated into the following: "Most of the rivers are regulated in some ways, mainly for hydropower and flood mitigation, but regulation generally consists of reservoirs with a simple weir

at their outlet (i.e., static control). Therefore, this did not prevent lumped models from reaching good performances in the former GRIP-O study of Gaborit et al. (in Press). As a consequence, no effort was made to represent in a detailed manner the artificial structures of the region in WATROUTE." 7) P.7, l.6, l.9: Which hydraulic parameters? Is the maximum soil depth calibrated for each grid cell or entire domain? AR: In Table 3, GEM-Hydro hydraulic parameters for the soil consist of GRKM, KVMO, PSMO, BMOD, WMOD, and to a lesser significant degree, RTD (see definitions in Table 3). The Maximum soil depth is currently fixed over the whole domain in GEM-Hydro. It is different for each of the 5 GRUs in SA-MESH. To make it clearer, the following sentence was modified into this: "The maximum soil depth is calibrated in GEM-Hydro and SA-MESH (Table 3 to Table 5). However, GEM-Hydro relies on a constant soil depth for the entire model, while SA-MESH uses a different soil depth value for each of its five GRUs."

8) P.7, l.19: what is RDPS? AR: It is the Regional Deterministic Prediction System, as defined close to page 7, line 14. 9) P8, l.3: how many runs has typically local calibration? AR: we used 400 with GEM-Hydro-UH, as mentioned close to page 8, line 17. The following sentence was extended as follows: "Calibration cost did not allow models to be calibrated locally for all GRIP-O subbasins (Fig. 1). One local calibration takes between 2 and 5 days of computation (400 model runs, see below)." 10) P.8, l.14: Please be more specific how were the values constrained? AR: the corresponding sentence was modified this way: " More precisely, soil water content thresholds and albedo (Table 3) cannot be higher than 1. Therefore, these values were constrained to realistic ranges after they were adjusted by the calibration algorithm by imposing them a minimum of 0 and a maximum of 1." 11) P.9, l.1-6: This part is not clear. On how many points is then the model verified/ compared /calibrated? AR: The model is verified against observed streamflows for tributaries entering Lake Ontario, at the location of the "most-downstream" gauge stations. In the case of subbasins having only one manor trbutary entering the Lake, the model was verified against the true observations of the most-downstream gauge of this tributary. But in the case of subbasins having more than one main tributary entering the Lake, all most-downstream observed

flows were summed up to end up with a unique time-series of observed flows, as if the catchment had only one major tributary. This was done in order to simplify the implementation of the lumped models of the first GRIP-O study, which cannot simulate streamflow at several points at a time. Therefore, lumped models were implemented on subbasins with more than one major tributary as if it had only one. Moreover, in such a case was this synthetic observed time-series extrapolated to the whole sub-basin area, including its ungauged areas, using a simple area-ratio method (ARM). The same framework was followed with GEM-Hydro UH for this type of subbasins for consistency with the lumped models (i.e., in order to be able to compare results of the different models in a fair manner), and because GEM-Hydro UH also can only simulate streamflow at one point at a time. This methodology is clearly explained in the first GRIP-O study cited in the paper, and is here briefly explained. However, a clarification effort was made and results in commented paragraph ending up as follows: " Finally, some subbasins in Fig. 1 have more than one major tributary flowing into Lake Ontario. In this case, the most-downstream observed flows on independent tributaries are summed and then extrapolated to the whole subbasin using the Area Ratio Method (ARM; Fry et al., 2014). The resulting "synthetic" flows were considered as observations for GEM-Hydro-UH calibration over the whole subbasin. This methodology was applied to all subbasins with more than one most-downstream gauge (identified with the "N/A" mention for the station attribute in Table 6) for consistency with the calibration experiments performed in the first GRIP-O study (see Gaborit et al., in Press), and because lumped models (and GEM-Hydro-UH) can only estimate streamflow at one location. For these subbasins, the true gauged fraction is specified in Table 6." 12) Strategy for ungauged basins: Typically, the prediction in ungauged basins is verified by leave-one-out approach. How do the results compare with such method? Please consider to discuss. AR:This method would have been too costly in terms of computational time given that it would require a number of global calibrations equal to the number of GRIP-O subbasins (i.e., 14). Given that it takes between 10 and 14 days for each global calibration to be performed, it can easily be noticed that the time required

for using the "leave-one-out" approach would be too long in this case. The "leave-one-out" methodology is generally performed when accompanied with local calibration which, despite requiring a distinct calibration for each of the subbasins, which is also very time-consuming, is much less costly than when using global calibration (local calibration takes about 4 days per subbasin to complete). In our case, we therefore use a completely different approach than those generally described in the Litterature for example by Razavi and Coulibaly (2012), and Parajka et al. (2013). The "leave-one-out" method does make sense when there's a high variability of the parameter values (as is the case with local calibration), which can not only be due to the model and the calibration algorithm themselves, but also to a high variability in catchment characteristics such as elevation, geology, land-use, soil texture, etc. Here, we have neither of these drawbacks: because we use global calibration, we end up with a unique, spatially consistent parameter set, and the area under study is relatively homogenous in terms of the aforementioned characteristics. Of course, a truly objective assessment of the parameter transfer proposed here would indeed require the "leave-one-out" approach, but this is not the objective of the study, which is instead of proposing a reliable and efficient manner of estimating streamflows for the ungauged areas of the watershed. The reliability of the method employed here (global calibration) is demonstrated by the fact that with a unique parameter set, the simulation performances are satisfying for all of the subbasins, which therefore logically leads to the assumption that it is satisfying for the ungauged areas as well, based on spatial and hydrologic proximity of the catchments. Moreover, temporal robustness of the method is also very good. Therefore, it is argued that the "leave-one-out" approach is not required in our case study to demonstrate the reliability of global calibration and parameter transfer to nearby ungauged catchments. In order to include these considerations in the paper, the following paragraph was added in section 2.3: " Despite a comprehensive assessment of the reliability of the methodology used here for parameter transfer would require the "leave-one-out" framework (see Razavi and Coulibaly, 2012), the satisfying performances and temporal robustness obtained for all GRIP-O subbasins with global calibration, along

with the spatial consistency of the unique final parameter set, the homogeneity of the area under study and the spatial proximity of ungauged catchments together justify the relevance and a priori reliability of the methodology employed in this study. This statement if moreover supported by the evaluation performed further down for the whole watershed."

13) P.11, l.28-29: Please consider to show some results supporting this statement. AR: this was done with the addition of the following paragraph along with the addition of the reference to NOHRSC (2004): " Calibration also improves GEM-Hydro-UH Snow Water Equivalent (SWE) simulations but to a lesser degree than for the streamflow. For example, the NSE values for SWE simulations over the 4 consecutive winters of the GRIP-O period improved from -0.12 to 0.42 for the Genessee subbasin, and from 0.49 to 0.68 for the Black River subbasin, respectively before and after calibration (the SWE variable was not used in the computation of the objective function). SWE observations come from the SNow Data Assimilation System (SNODAS, see NOHRSC 2004)." 14) P.16, l.17: Please consider to update XXXX. AR: this will be done at the final stage of the publication process.  15) P.21, l.5: "the most-downstream flow gauges" is not clear.  AR: This legend was updated as follows: " Dots (blue for natural flow regimes and red for regulated regimes) are the most-downstream flow gauges (i.e., the main tributaries' gauges which are closest to Lake Ontario's shoreline) selected for model calibrations. " 16) Table 1: Please be more specific what is radiative forcings AR: it was specified as follows: " R:, radiative forcings (short- and long-wave incoming radiations)"

Please also note the supplement to this comment:
http://www.hydrol-earth-syst-sci-discuss.net/hess-2016-508/hess-2016-508-SC1-supplement.zip

---

## Referee Comment (RC2) · Anonymous Referee #2 · 24 Jan 2017

*Hydrol. Earth Syst. Sci. Discuss.*

Manuscript Number: doi:10.5194/hess-2016-508, 2016

Title: A Hydrological Prediction System Based on the SVS Land-Surface Scheme: Implementation and Evaluation of the GEM-Hydro platform on the watershed of Lake Ontario

**General comments**

This is an interesting paper describing the development of a modelling system to estimate net basin supply to a strategic Canada/USA water body: Lake Ontario. From a technological point of view, the work is state of the art and clearly highlights recent efforts undertaken by Environment and Climate Change Canada to develop a robust hydrological modelling system. More specifically, the objectives of this paper are to:

(i)    ''propose a methodology for calibrating the distributed GEM-Hydro platform developed by ECCC in order to improve streamflow simulations for Lake Ontario, which we expect would ultimately propagate into improved simulations of Lake Ontario Net Basin Supplies (or NBS, the sum of lake tributary runoff, overlake precipitation, and overlake evaporation: Brinkmann 1983);

(ii)   compare GEM-Hydro with two other distributed models (inter-comparison study) in order to identify avenues to further improve GEM-Hydro; and

(iii)  propose and evaluate a method for estimating runoff for the ungauged parts of the watershed.''

Objective (i)

As far as I can get, the paper never actually demonstrated what is highlighted in yellow. So unless the reader is an insider, there is no evidence that currently applied modelling systems do not provide satisfactorily the sought-after streamflow simulations. I understand with the interest of developing a state-of-the-art modelling system, but the paper does not provide a strong motivation. The authors need to convince the readers here.

Objective (ii)

After reading the paper a couple of times, I feel the paper has yet to actually and clearly identify the avenues to further improve GEM-Hydro. The authors mentioned that SVS would benefit from a soil heat balance equation based on one missed spring peakflow; that is farfetched as it could have resulted from multiple sources, so I believe the paper does not make a strong case here. Now, I am not sure the reader will get anything out of this objective and, at best, I think the model intercomparison should be considered as supplemental material. Furthermore, neither the abstract, nor the conclusion, provide any tangible answers to this objective.

The above comments might seem harsh, but they are factual. Do not get me wrong I think the paper reports relevant technological information to the hydrometeorological community. The paper shows promises and I believe the authors can streamline the content to the essentials, that is the demonstration of the advantage of substituting a unit hydrograph for WATROUTE during calibration can actually reduce the computational time required for model calibration and illustrate that GEM-Hydro can benefit from a local/global calibration strategy to provide '' good streamflow predictions ''. Should the authors decide to follow this suggestion, they should substantiate their rationale from a scientific point of view rather than a technological one. For example, they should provide more fundamental information between the computational time scales of the LSS and those of WATROUTE and UH. Furthermore, they should discuss the relationship between the computational time scales and the dimension of the computational elements used in WATROUTE and the UH versus those used in the LSS.

It is noteworthy, Section 1.4. is confusing. In fact, although I am somewhat familiar with the work of the first author on local and global calibration, but I needed to read the section more than once to comprehend; and even then, I am still not sure what was actually done. I do not quite follow the use of GR4J in the calibration of GEM-Hydro-UH (see my comment below related to the content of P.10, lines 15-17) because the paper does not provide the underlying hypothesis. I strongly recommend the authors to provide a diagram or sketch describing the steps taken to achieve the calibration strategy introduced in Section 1.4. At this point, I doubt that most readers can appreciate what the authors actually did.

I have made suggestions in the following list of specific comments below on how to fulfill what I perceive as shortcomings. I have also added a few editorial comments to improve the paper. As a side note, I found a bit difficult the exercise of going back and forth between the content of the introductory sections to remind myself what the acronyms meant. The authors should provide a list of acronyms to facilitate the reading of the manuscript.

I strongly encourage the authors to address these comments as I feel the paper could certainly be a good technological contribution to the hydrometeorological community.

**Specific comments**

- P. 2, line 10: the following sentence:
  - « Going beyond anomaly forecasts (which are bias corrected based on a model climatology) to obtain unbiased short-term streamflow forecasts is more challenging due to limitations of operational Land-Surface Schemes (LSS), which are generally geared towards improving weather forecasts, sometimes at the cost of not representing (or misrepresenting) surface and subsurface hydrological processes that are critical to hydrological simulation. »
    - …should be modified as follows:

- Going beyond anomaly forecasts, which are bias-corrected based on a modeled climatology to obtain unbiased short-term streamflow forecasts, is more challenging. This is due to limitations of operational Land-Surface Schemes (LSS), which are generally geared towards improving weather forecasts, sometimes at the cost of not representing (or misrepresenting) critical surface and subsurface hydrological processes.
- P.2, line 15: the following sentences:
  - Hydrological processes in land-surface models used for NWP are improving quickly (Balsamo et al., 2009; Masson et al., 2013; Alavi et al., 2016; Wagner et al., 2016), as soil water content and snow are recognized as important sources of their predictability that remain to be fully tapped into (Koster et al., 2004; Entekhabi et al., 2010). Environment and Climate Change Canada (ECCC), the Canadian department that provides operational weather and environmental forecasts, is in the process of implementing a major upgrade to the LSS used by its NWP model, the Global Environmental Multi-scale model (GEM)
    - …should be modified as follows:
      - Hydrological processes simulated by land-surface schemes (LSS) used for NWP are improving quickly (Balsamo et al., 2009; Masson et al., 2013; Alavi et al., 2016; Wagner et al., 2016), as soil water content and snow water equivalent are recognized as key state variables for streamflow forecasting (Koster et al., 2004; Entekhabi et al., 2010). Environment and Climate Change Canada (ECCC), which provides operational weather and environmental forecasts within its boundary, is in the process of implementing a major upgrade to the LSS of the Global Environmental Multi-scale model (GEM), the national model.
- P. 2, line 20: please delete « …in order… »
- P.3, line 6: please modify as follows:  « …thermodynamics, as reported by Wiley… »
- P.5, line 6: please correct me if I am wrong, but WATFLOOD has no LSS, just a simple potential evapotranspiration equation, unless WATCLASS was used.  So WATFLOOD is more along the line of GR4J with that respect.
- P.5, lines 5 through 34, I think there is room here to provide more fundamental information between the computational time scales of the LSS and those of WATROUTE and UH.
  - Furthermore, discuss the relationship between the computational time scales and the dimension of the computational elements used in WATROUTE and the UH.
- P.6, line 10: what does SA mean in SA-MESH?
- P.6, line 21: please replace «… the outlet of Lake Ontario.» by «..the Lake outlet. »

- P.7, line 10: please replace « … that it is higher than 1 m.. » by « … that it is greater than 1 m.. »
- P.7, line 20: is there any spin-up for the calibration period?
- P.7, equation (1): why presenting the PBIAS expression and not the NS...the latter being more complex than the former…
- P.8, line 3:
    - Please specify those GEM-Hydro-UH GRIP-O sub-basins that were locally calibrated out of all sub-basins?
        - What is the the percentage of the Lake Ontario basin that had local model calibration? Is it 88.5% (P.9, line 15) ?
- P.9, line 1: «…some subbasins in Fig. 1 have several gauge stations. »
    - It would help if these gauge stations could be displayed, but I assume it might not be feasible given the coarse resolution of this figure.
- P.10, lines 1-9: the equifinality problem still exists for the global calibration, please discuss?
- P.10, line 10: What does a unique implementation mean? It is not clear. I assume GR4J was first calibrated on each gauged sub-basin, then the global calibration took place and a single parameter set was found. Please define unique.
- P.10, line 16: What do you mean here: « …performances obtained with local Gr4J calibrations (Gaborit et al., in Press) were used when needed…».
    - Do you mean that for those sub-basins not modelled by GEM-Hydro-UH, the performances of GR4J were substituted in the computation of the objective function (Eq. 2)?
    - The hypothesis behind this approach must be clearly stated in the paper; that is it is assumed that GEM-Hydro-UH would have a similar performance, am I right?
- P.10, lines 15-17: « However, as GEM-Hydro-UH was not locally calibrated for all of the 14 GRIP-O subbasins, performances obtained with local GR4J calibrations (Gaborit et al., in Press) were used when needed (justifying the use of that model in this study). »
    - How was this done? Please provide a quick summary so the reader doesn't have to access the reference.
- P.10, line 21: it is not arbitrary if it is based on prior work!
- P.10, line 32 & P.11, lines 1 & 12: Watroute should be written with capital letters (WATROUTE).
- P.11, line 23: replace « …which…» by « …whose…»
- P.13, line 8: Please be consistent and replace watershed by basin.
- P.15, lines 20-21: « However, as a limited number of subbasins were used for the inter-comparison due to computational time limitations, no general model ranking can be derived from this study. ».
    - This means perhaps this paper is premature. Or as mentioned in the general comment section. Model intercomparison should be considered as supplemental information.

- P.16, lines 4-5: I still do not get it, perhaps WATROUTE needs to be calibrated separately otherwise why calibrating with the UH? It is only valid to use WATROUTE if it can reproduce the UH at the chosen outlets used for the UH calibration. Unless there is a philosophical point I am not getting, which is perhaps possible, but doubtful. Please make a strong rebuttal to this statement.
- P.16, line 17: please replace No. XXXX
- P. 17- 20: In the References section, there are several references with « …, », please fill them in.

**Figures**

- Figure1
    - The word « areas » should be replaced by sub-bassins, drainage areas or basins.
    - In the figure caption, replace sub-catchment by sub-basin, please be consistent.
- Figure 2
    - Moira river (CA) should be replaced by Moira River (CAN).  CA usually stands for California.
    - Please remind the reader that the Moira River basin is sub-basin 11.
- Figure 3
    - Correct me if I am mistaken, but shouldn't the caption be as follows: Uncalibrated GEM-Hydro and GEM Hydro-UH performances…
    - Wouldn't be interesting to discuss the differences, at least for one or two sub-basins?
- Figure 4
    - Replace sub-catchment by sub-basin, please be consistent.
- Figure 5
    - Please use upper case letters for Mesh (MESH), Watflood (WATFLOOD), please be consistent.
- Figure 6
    - Why are not there any local calibration for sub-basins 13, 14 and 15
    - What are the default parameter values when compared to those resulting from the calibration procedure, local and global? Wouldn't be interesting to discuss the differences, at least for one or two sub-basins?
    - Please add the following precisions to the figure caption (at least that is my assumption):  Results are presented as NSE √ (left) and PBIAS (right), for many GRIP-O sub-basins.
- Figure 8
    - Cumulative monthly NBS cannot by definition flow rate units, the units here should be cubic meters.
    - What are the numbers 7, 11 and 3 on the x-axis? Sub-basins number? I assume so as there are 14 tick marks between the occurrences of the number 7. Please provide this information in the figure caption.

**Tables**

- Table 4
    - The range for some parameter values defies the imagination, or any explanations?
- Table 6
    - To avoid any confusion please substitute CA for CAN, which is more often used - otherwise CA usually refers to California

**Answer to traditional questions**

Is the paper free of errors in logic?

- Yes

Do the conclusions follow from the evidence?

- Yes and no – see general comments.

Are alternative explanations explored as appropriate?

- Yes.

Are biases, limitations, and assumptions clearly stated, and uncertainty quantified?

- Yes.

Is methodology explained in sufficient detail so that the paper's scientific conclusions could be tested by others?

- No, see the above list of general comments.

Is previous work and current understanding cited and represented correctly?

- No, see the above list of general comments about local and global calibration strategy.

Is information conveyed clearly enough to be understood by the typical reader?

- Yes and no – see general comment about local and global calibration strategy

Are all figures and tables necessary, appropriate, legible, and annotated (as appropriate)?

- Yes and no – see aforementioned comments

---

## Author Comment (AC1) · 8 Feb 2017

Answers to the reviewer follow the "AR" letters.

Hydrol. Earth Syst. Sci. Discuss. Manuscript Number: doi:10.5194/hess-2016-508, 2016 Title: A Hydrological Prediction System Based on the SVS Land-Surface Scheme: Implementation and Evaluation of the GEM-Hydro platform on the watershed of Lake Ontario General comments This is an interesting paper describing the development of a modelling system to estimate net basin supply to a strategic Canada/USA water body: Lake Ontario. From a technological point of view, the work is state of the art and clearly highlights recent efforts undertaken by Environment and Climate Change Canada to develop a robust hydrological modelling system. More specifically, the objectives of this paper are to: (i) ''propose a methodology for calibrating the distributed GEM-Hydro platform developed by ECCC in order to improve streamflow simulations for Lake Ontario, which we expect would ultimately propagate into improved simulations of Lake Ontario Net Basin Supplies (or NBS, the sum of lake tributary runoff, overlake precipitation, and overlake evaporation: Brinkmann 1983); (ii) compare GEM-Hydro with two other distributed models (inter-comparison study) in order to identify avenues to further improve GEM-Hydro; and (iii) propose and evaluate a method for estimating runoff for the ungauged parts of the watershed.''

Objective (i) As far as I can get, the paper never actually demonstrated what is highlighted in yellow. So unless the reader is an insider, there is no evidence that currently applied modelling systems do not provide satisfactorily the sought-after streamflow simulations. I understand with the interest of developing a state-of-the-art modelling system, but the paper does not provide a strong motivation. The authors need to convince the readers here.

AR: after the answers to the first reviewer, the sentence highlighted in yellow was removed, and the first objective was reformulated as follows: " this study mainly aims at finding a methodology to implement the distributed GEM-Hydro model over the whole Lake Ontario watershed, including its ungauged parts, in an efficient manner. Distributed models are more complicated to implement and more computationally-intensive than lumped ones, but have a broader range of applications." However, it is true that this work did improve streamflow simulations in comparison to former studies, which mainly consist of those of Croley and Haghnegahdar, as mentioned in the introduction of the first GRIP-O paper (Gaborit et al., 2016 a): "In the Lake Ontario watershed, Croley (1983) and Haghnegahdar et al. (2014) calibrated hydrologic models with

runoff observations to optimize simulations of this variable. The former demonstrated that the LBRM worked well for simulating weekly flows, while the second evaluated the MESH model on 15 Great Lakes subbasins, including two Lake Ontario subbasins. However, even after calibration, the MESH model did not perform particularly well for validation catchments (Nash-Sutcliffe values below 0.6)." Therefore, an additional sentence was added in results section, close to p.12, l.10: "Therefore, GRIP-O allowed to improve streamflow simulations for the Lake Ontario basin, in comparison to the studies of Croley (1983) and Haghnegahdar et al. (2014), which are the main former studies who proposed the implementation of hydrologic models over this area."

Objective (ii) After reading the paper a couple of times, I feel the paper has yet to actually and clearly identify the avenues to further improve GEM-Hydro. The authors mentioned that SVS would benefit from a soil heat balance equation based on one missed spring peakflow; that is farfetched as it could have resulted from multiple sources, so I believe the paper does not make a strong case here. Now, I am not sure the reader will get anything out of this objective and, at best, I think the model intercomparison should be considered as supplemental material. Furthermore, neither the abstract, nor the conclusion, provide any tangible answers to this objective.

AR: We do not agree with the reviewer that this objective is useless despite we have only results for 3 subbasins for the intercomparison of the models. As clearly stated in the text, this inter-comparison allows to emphasize these important facts: "Even if the intercomparison is obviously limited in the number of available test cases, it allows highlighting the mandatory need of calibrating hydrologic models, that models have unique behaviors that translate in substantial differences in hydrographs, and that each of the models could benefit from some strengths of its competitors". It does also show that GEM-Hydro is a priori competitive with the two other main distributed models used in this region. In regard of the avenues for improving GEM-Hydro, we do not claim that the difference between the model simulations is due to the lack of soil freezing and melting processes in SVS, as can be seen in the following sentence with the use "which may be

due to": "Peak flow events associated to the spring freshet are generally better represented by MESH, which may be due to a better representation of the soil freezing and melting processes occurring in CLASS (MESH LSS)." Moreover, we do not make this assumption based on one spring peak flow event, but we have three subbasins with 7 years of simulations, which does begin to be a significant number of events. Finally, it is true that SVS should benefit from the implementation of the soil freezing and melting processes, because it is the aim of this LSS to represent physical processes. The potential benefit of a more complex snow module is also straightforward in comparison to the current force-restore approach (which only relies on an average snowpack temperature to estimate snowmelt), given the complexity of the physical processes related to snow. Therefore, we do believe that these two avenues consist in some of the main ones to further improve SVS. We did add another avenue of improvement for SVS with the following sentence (p. 15, l.18, in the conclusion): "Finally, work is under way to represent a surface of variable area of ponded water in each SVS grid cell, in order to represent subgrid-scale lakes, wetlands, and to better represent the delay associated with surface runoff transfer into streams." 2 The above comments might seem harsh, but they are factual. Do not get me wrong I think the paper reports relevant technological information to the hydrometeorological community. The paper shows promises and I believe the authors can streamline the content to the essentials, that is the demonstration of the advantage of substituting a unit hydrograph for WATROUTE during calibration can actually reduce the computational time required for model calibration and illustrate that GEM-Hydro can benefit from a local/global calibration strategy to provide '' good streamflow predictions ''. Should the authors decide to follow this suggestion, they should substantiate their rationale from a scientific point of view rather than a technological one.

For example, they should provide more fundamental information between the computational time scales of the LSS and those of WATROUTE and UH. Furthermore, they should discuss the relationship between the computational time scales and the dimension of the computational elements used in WATROUTE and the UH versus those used

in the LSS.

AR: Information was added on the computational times required by the system. (see specific comment on p.5, lines 5-34).

It is noteworthy, Section 1.4. is confusing. In fact, although I am somewhat familiar with the work of the first author on local and global calibration, but I needed to read the section more than once to comprehend; and even then, I am still not sure what was actually done.

AR: we know this section is hard to follow but we did prefer explain clearly what was done rather than take the risk to leave the reader with unanswered questions regarding the methodology. Moreover, the methodology proposed also illustrates that different possibilities do exist to fulfill this kind of work, and that it is important to envision the different ones before starting the work, which is still an important message to remind to modelers.

I do not quite follow the use of GR4J in the calibration of GEM-Hydro-UH (see my comment below related to the content of P.10, lines 15-17) because the paper does not provide the underlying hypothesis.

AR: see answer to specific comment on page.10, line 16.

I strongly recommend the authors to provide a diagram or sketch describing the steps taken to achieve the calibration strategy introduced in Section 1.4. At this point, I doubt that most readers can appreciate what the authors actually did.

AR: we will let the publishing authority decide on this but we do believe that the text of section 1.4 accompanied with Fig. 1 does allow to understand.

I have made suggestions in the following list of specific comments below on how to fulfill what I perceive as shortcomings. I have also added a few editorial comments to improve the paper. As a side note, I found a bit difficult the exercise of going back and forth between the content of the introductory sections to remind myself what the

acronyms meant. The authors should provide a list of acronyms to facilitate the reading of the manuscript.

AR: A list of acronyms will be put in supplementary material.

I strongly encourage the authors to address these comments as I feel the paper could certainly be a good technological contribution to the hydrometeorological community. Specific comments • P. 2, line 10: the following sentence: o Âń Going beyond anomaly forecasts (which are bias corrected based on a model climatology) to obtain unbiased short-term streamflow forecasts is more challenging due to limitations of operational Land-Surface Schemes (LSS), which are generally geared towards improving weather forecasts, sometimes at the cost of not representing (or misrepresenting) surface and subsurface hydrological processes that are critical to hydrological simulation. Âż . . .should be modified as follows:

• Going beyond anomaly forecasts, which are bias-corrected based on a modeled climatology to obtain unbiased short-term streamflow forecasts, is more challenging. This is due to limitations of operational Land-Surface Schemes (LSS), which are generally geared towards improving weather forecasts, sometimes at the cost of not representing (or misrepresenting) critical surface and subsurface hydrological processes.

AR: this sentence was removed from the text due to reviewer 1 comments who suggested to shorten the introduction.

• P.2, line 15: the following sentences: o Hydrological processes in land-surface models used for NWP are improving quickly (Balsamo et al., 2009; Masson et al., 2013; Alavi et al., 2016; Wagner et al., 2016), as soil water content and snow are recognized as important sources of their predictability that remain to be fully tapped into (Koster et al., 2004; Entekhabi et al., 2010). Environment and Climate Change Canada (ECCC), the Canadian department that provides operational weather and environmental forecasts, is in the process of implementing a major upgrade to the LSS used by its NWP model, the Global Environmental Multi-scale model (GEM) . . .should be

modified as follows: • Hydrological processes simulated by land-surface schemes (LSS) used for NWP are improving quickly (Balsamo et al., 2009; Masson et al., 2013; Alavi et al., 2016; Wagner et al., 2016), as soil water content and snow water equivalent are recognized as key state variables for streamflow forecasting (Koster et al., 2004; Entekhabi et al., 2010). Environment and Climate Change Canada (ECCC), which provides operational weather and environmental forecasts within its boundary, is in the process of implementing a major upgrade to the LSS of the Global Environmental Multi-scale model (GEM), the national model.

AR: this was done.

• P. 2, line 20: please delete Âń . . .in order. . . Âż

AR: done

• P.3, line 6: please modify as follows: Âń . . .thermodynamics, as reported by Wiley. . . Âż

AR: done

• P.5, line 6: please correct me if I am wrong, but WATFLOOD has no LSS, just a simple potential evapotranspiration equation, unless WATCLASS was used. So WATFLOOD is more along the line of GR4J with that respect.

AR: Yes, but it is still distributed and in this sense is very different from GR4J. However, the sentence below was added close to page 5, line 8: "It relies on the GRUs concept and on many empirical equations."

• P.5, lines 5 through 34, I think there is room here to provide more fundamental information between the computational time scales of the LSS and those of WATROUTE and UH. o Furthermore, discuss the relationship between the computational time scales and the dimension of the computational elements used in WATROUTE and the UH.

AR: the sentences below were elongated at page 5, line 13 and page 5, line 23: " Sensitivity tests (Gaborit et al., 2016 b) revealed that 2 and 10 arcmin resolutions for SVS lead to quite similar performance in terms of streamflow at the outlet, while a substantial amount of computational time is saved when running the coarser resolution (almost proportionally if using the same number of nodes)." "As the GEM-Hydro suite (including WATROUTE) is quite demanding in terms of computational time, it was decided to test a stand-alone configuration of GEM-Hydro relying on text files only and in which WATROUTE is replaced by a Unit Hydrograph (UH).This version is here forth referred to GEM-Hydro-UH"

And the whole following paragraph was added close to page 5, line 23:

" Indeed, the computational time for the experiment setup described here and when splitting the domain in four on an ECCC supercomputer is about 1.5 min per day for the LSS part of GEM-Hydro (SVS), provided that the pre-processing of the atmospheric variables was already done (which is the case in calibration: the pre-processing is done only once). The WATROUTE code is not yet parallelized, each grid point being processed from upstream to downstream, but requires only 25s per day for the setup described here when running on a local machine. However, the WATROUTE pre-processing (i.e., preparation of the WATROUTE input files from the SVS outputs) is very long - about 30s per day. Therefore, WATROUTE computational time was still lower than SVS one for this setup and was not the limiting factor. One simulation run over the GRIP-O period (4.5 years) therefore requires about 1.7 days with GEM-Hydro and prevents from performing any automatic calibration (which requires at least 400 runs, see below). Instead of using GEM-Hydro to run SVS, a stand-alone SVS version, coded in Fortran, was used. This executable saves a tremendous amount of computation time compared to GEM-Hydro mainly because of the Input/Output processing time: the stand-alone version makes use of text files which are kept open during the simulation and requires only 4.5s per day on a local machine for this setup (2 h for the 4.5 years GRIP-O period or 30 days of calibration with 400 runs if running the

whole domain). However, the computational time required by WATROUTE still had to be bypassed to perform automatic calibrations, which was done with the UH concept."

• P.6, line 10: what does SA mean in SA-MESH?

AR: Stand-alone, but it was removed for consistency throughout the paper.

• P.6, line 21: please replace Âń. . . the outlet of Lake Ontario.Âż by Âń..the Lake outlet. Âż AR: done 4 • P.7, line 10: please replace Âń . . . that it is higher than 1 m.. Âż by Âń . . . that it is greater than 1 m.. Âż

AR: done

• P.7, line 20: is there any spin-up for the calibration period?

AR: p. 7 line 24, it was specified that yes, we do use spin-up for the calibration. " Validation is from June 1st, 2005 to June 1st, 2007 (2 years, last one being used as spin-up for calibration)" and right after: " Note that during the automatic calibrations, the spin-up year was simulated only once and for all subsequent runs."

• P.7, equation (1): why presenting the PBIAS expression and not the NS...the latter being more complex than the former. . .

AR: but way more used in the hydrologic community than this "percent" normalized BIAS criteria.

• P.8, line 3: o Please specify those GEM-Hydro-UH GRIP-O sub-basins that were locally calibrated out of all sub-basins? What is the the percentage of the Lake Ontario basin that had local model calibration? Is it 88.5% (P.9, line 15) ?

AR: it was emphasized that local calibration was not achieved for all sub-basins "but only those shown on Fig. 3" at this location.

• P.9, line 1: Âń. . .some subbasins in Fig. 1 have several gauge stations. Âż o It would help if these gauge stations could be displayed, but I assume it might not be

feasible given the coarse resolution of this figure.

AR: they are, with the blue or red circles.

 c P.10, lines 1-9: the equifinality problem still exists for the global calibration, please discuss?

AR: the following sentence was added to emphasize this fact: " Despite global calibration may not be exempt of equifinality, the attention paid to the parameter ranges used (Table 3) allows to be confident in the physical relevance of the final parameter values."

 c P.10, line 10: What does a unique implementation mean? It is not clear. I assume GR4J was first calibrated on each gauged sub-basin, then the global calibration took place and a single parameter set was found. Please define unique.

AR: it is already defined at page 9, line 22, as the synonym of "single". It cannot be clearer. One model for one area, either the total gauged area or the whole area, including ungauged portions. However, it was specified at p.10, line 10, to "(see above for the justification)" of this methodology, because as explained above, the unique model leads to similar performances than those obtained with several local models.

 c P.10, line 16: What do you mean here: Âń . . .performances obtained with local Gr4J calibrations (Gaborit et al., in Press) were used when needed. . .Âż. o Do you mean that for those sub-basins not modelled by GEM-Hydro-UH, the performances of GR4J were substituted in the computation of the objective function (Eq. 2)? AR: Yes, and this sentence was modified as follows to be clearer: " However, as GEM-Hydro-UH was not locally calibrated for all of the 14 GRIP-O sub-basins (only those of Fig. 3 because of the computation cost), performances obtained with local GR4J calibrations (Gaborit et al., 2016 a) were used for the remaining ones to set the reference performance to be used in Eq. (2), justifying the use of that model in this study."

o The hypothesis behind this approach must be clearly stated in the paper; that is it is assumed that GEM-Hydro-UH would have a similar performance, am I right?

[Figure]

AR: the hypothesis is more that the local GR4J performance consists in the maximum performance which can be reached. Although mentioned under a more temperate statement, this was clarified in the Paper:

"This substitution does make sense considering that firstly, GR4J and GEM-Hydro-UH local performances are similar (Fig. 3), secondly that GR4J local performances were always very satisfactory (see Gaborit et al., 2016 a), and thirdly that the objective function still makes sense if global performance is higher than the local one (see above)." It seemed important to the authors' eyes to emphasize the last point of the above sentence by added, just above it, the following one: "It [the objective function] does rely on the hypothesis that global performance cannot be higher than local performance, but even if it was the case, this objective function would still make sense and the gain achieved with global over local performance would simply compensate for errors obtained on other catchments, possibly allowing to reach a perfect objective function value even with catchments having poorer performances with global calibration than with local calibration."

• P.10, lines 15-17: Âń However, as GEM-Hydro-UH was not locally calibrated for all of the 14 GRIP-O subbasins, performances obtained with local GR4J calibrations (Gaborit et al., in Press) were used when needed (justifying the use of that model in this study). Âż o How was this done? Please provide a quick summary so the reader doesn't have to access the reference.

AR: given the clarifications brought in the previous answer to the reviewer, it is believed this point is clarified and does not need further explanation.

• P.10, line 21: it is not arbitrary if it is based on prior work!

AR: this was corrected.

• P.10, line 32 & P.11, lines 1 & 12: Watroute should be written with capital letters (WATROUTE).

AR: done

• P.11, line 23: replace Âń . . .which. . .Âż by Âń . . .whose. . .Âż

AR: done

• P.13, line 8: Please be consistent and replace watershed by basin.

AR: done

• P.15, lines 20-21: Âń However, as a limited number of subbasins were used for the inter-comparison due to computational time limitations, no general model ranking can be derived from this study. Âż. o This means perhaps this paper is premature. Or as mentioned in the general comment section. Model intercomparison should be considered as supplemental information.

AR: see answer to the general comments

P.16, lines 4-5: I still do not get it, perhaps WATROUTE needs to be calibrated separately otherwise why calibrating with the UH? It is only valid to use WATROUTE if it can reproduce the UH at the chosen outlets used for the UH calibration. Unless there is a philosophical point I am not getting, which is perhaps possible, but doubtful. Please make a strong rebuttal to this statement.

AR: as clarified when answering the comment on P.5, lines 5 through 34 by giving more details on the computational time requirements, replacing WATROUTE with a simple UH was mandatory to proceed with the automatic calibrations; it is true that when the LSS is calibrated, we then could calibrate WATROUTE parameters using the calibrated SVS parameters in order to truly get the maximum possible performances. This was done in the GRIP-O report cited in the paper: Gaborit et al., 2016b. However, this requires a long calibration time for only 4 parameters. Moreover, experience has shown that a manual adjustment of these 4 parameter values allows to achieve simulation performances which are close to those obtained after automatic calibration. (The two LZS parameters are adjusted maximizing the recession and low-flow period simulations

while the two Manning coefficients control the magnitude of peak flow events). So this manual work is quite straightforward with some training, and what is sure is that if the UH can do it, WATROUTE can do it. So the target when tuning WATROUTE parameters may even be the UH simulations themselves rather than the observations, because the calibrated UH simulations consist in the optimal simulations for a given methodology.

This was emphasized by slightly extending the following sentence (page 16 line 4): " The routing part of GEM-Hydro can be run afterwards, potentially adjusting the standard Manning values if needed (which can be done manually with a few runs)"

• P.16, line 17: please replace No. XXXX

AR: this will be done right before the final publication.

• P. 17- 20: In the References section, there are several references with Âń . . ., Âż, please fill them in.

AR: done

Figures • Figure1 o The word Âń areas Âż should be replaced by sub-bassins, drainage areas or basins. o In the figure caption, replace sub-catchment by sub-basin, please be consistent.

AR: done for sub-catchment, but the term "areas" does not designate any subbasin but encompasses parts of several subbasins and therefore needs to remain different from the term subbasin.

• Figure 2 o Moira river (CA) should be replaced by Moira River (CAN). CA usually stands for California. o Please remind the reader that the Moira River basin is subbasin 11.

AR: done

• Figure 3 o Correct me if I am mistaken, but shouldn't the caption be as follows: Uncalibrated GEM-Hydro and GEM Hydro-UH performances. . .

AR: No, results only show GEM-Hydro-UH performances, before and after calibration.

o Wouldn't be interesting to discuss the differences, at least for one or two sub-basins?

AR: It is not believed that more information is required for Figure 3 than its current description (page 11, line 24).

⇢ Figure 4 o Replace sub-catchment by sub-basin, please be consistent.

AR: done

⇢ Figure 5 o Please use upper case letters for Mesh (MESH), Watflood (WAT-FLOOD), please be consistent.

AR: done

⇢ Figure 6 o Why are not there any local calibration for sub-basins 13, 14 and 15 AR: because of computation time; the sentence close to p.10, l.15 was modified into: " However, as GEM-Hydro-UH was not locally calibrated for all of the 14 GRIP-O sub-basins (only those of Fig. 3 because of the computation cost)"

o What are the default parameter values when compared to those resulting from the calibration procedure, local and global? Wouldn't be interesting to discuss the differences, at least for one or two sub-basins? AR: Table 7 gives some info. about parameter values; for the sake of brevity it is not believed that a more precise comparison between parameter values obtained with local and global calibrations is needed for a particular catchment, the main conclusion being that there is a strong variability in parameter values obtained with local calibration, which is shown in Table 7. However, the following sentence was added close to p.13, l. 11: " Moreover, it was noticed (not shown here) that parameter values were very different between local and global calibration procedures, even for catchments displaying very similar performances between the two strategies (such as subbasins 3, 5 and 8, see Fig. 6)."

o Please add the following precisions to the figure caption (at least that is my assump-
tion): Results are presented as NSE $\sqrt{}$ (left) and PBIAS (right), for many GRIP-O sub-basins.

AR: done

• Figure 8 o Cumulative monthly NBS cannot by definition flow rate units, the units here should be cubic meters.

AR: done

o What are the numbers 7, 11 and 3 on the x-axis? Sub-basins number? I assume so as there are 14 tick marks between the occurrences of the number 7. Please provide this information in the figure caption.

AR: they are months. This was specified. 6 Tables • Table 4 o The range for some parameter values defies the imagination, or any explanations? AR: No, the calibration was performed outside of ECCC.

• Table 6 o To avoid any confusion please substitute CA for CAN, which is more often used - otherwise CA usually refers to California

AR: done

Please also note the supplement to this comment:
http://www.hydrol-earth-syst-sci-discuss.net/hess-2016-508/hess-2016-508-AC1-supplement.zip

---

## Author Response (AR1)

**Editor Decision: Reconsider after major revisions (further review by Editor and Referees)**
(14 Feb 2017) by Wouter Buytaert
Comments to the Author:
Dear Dr. Gaborit,

thank you for replying to the comments of the two reviewers. From those, you will have noted that both reviewers find the presented results potentially interesting, and suggest that the topic may be suitable for publication in HESS.

However, you will also have noticed that both reviewers have serious concerns about the clarity of writing and overall quality of presentation of the manuscript. Your replies to the reviewers address these concerns to some extent. But after having read carefully the revised manuscript that you attached to your reply to reviewer 2, I am concerned that this may not constitute the major revisions which the reviewers are seeking, and which I fully support.

One example is section 1.4: I agree with reviewer 2 that the calibration procedure presented here is very hard to follow and requires revision (and may well benefit from a diagram). For instance, when you mention that calibration was performed on "the GRIP-O gauged area", do you mean that the model was calibrated on each of the gauged subbasins individually, or some kind of aggregate? If the former, how were the parameters then transferred to the ungauged area; if the latter, how was the aggregate obtained? It is important to keep in mind that reproducibility is a corner stone of scientific publication. I do not think that this is already achieved in the version that you attached in your reply to reviewer 2, especially if one considers that a considerable number of readers of HESS may not be thoroughly familiar with the GEM-Hydro setup.

These considerations lead me to request major revisions. The manuscript will be sent back to the reviewers, with the specific request to evaluate whether their concerns about presentation and transparency of the model implementation and evaluation have been addressed adequately, in addition to any scientific concerns.

Such revisions would also give you the opportunity, as reviewer 1 suggests, to increase the scientific relevance and perhaps even the streamline some of the content.

Depending on the availability of the original reviewers, I may also seek the opinion of a third reviewer.

At this point, I cannot guarantee that your manuscript will eventually be published in HESS; this will entirely depend on the thoroughness with which you will be able to address the reviews, which I believe are very thorough and constructive.

Kind regards
Wouter Buytaert
handling editor

Answer to Editor:

The section 1.4 was reformulated and a diagram was added; A reference was added for the Unit Hydrograph (UH) in section 1.2; the following sentence was added in section 1.1: " The basin averages are computed as a weighted average of the SVS grid cells located in the considered basin."

the following sentence was added in section 2.1 to give another detail related to the interpretation of Fig. 4 (formerly Fig.3): " It can also be noticed on Fig. 4 that calibration sometimes inverts the sign of the

PBIAS criteria (switching from over- to under-estimation or vice-versa)." A sentence was also extended at the beginning of section 2.3 (extended part in red) in order to add some information about the parameter value differences between local and global calibration as suggested by reviewer 1: " Moreover, it was noticed (not shown here) that parameter values were very different between local and global calibration procedures, even for catchments displaying very similar performances between the two strategies (such as subbasins 3, 5 and 8, see Fig. 7), highlighting the fact that local calibration is more prone to over-calibration (i.e., equifinality)."

In order to mitigate the fact that we derive some hypotheses based on a few events only (as suggested by reviewer 2), the following sentence (in section 2.2):

" Peak flow events associated to the spring freshet are generally better represented by MESH, which may be due to a better representation of the soil freezing and melting processes occurring in CLASS (MESH LSS)."

was replaced with this one:

 "Peak flow events (even for other subbasins) associated to the spring freshet are generally better represented by MESH, which may be due to a better representation by CLASS of various cold regions hydrological processes, such as snow accumulation and melt, snow interception by vegetation, as well as soil freezing and thawing."

Finally, the manuscript was read in order to try to streamline the content where possible.

We hence believe that reviewer 2 comments have been mostly satisfied, except regarding the intercomparison section which we would prefer to let in the paper for the reasons mentioned in the answer to reviewer 2 comments.

[revised manuscript text omitted]

---

## Referee Report (RR1)

*Hydrol. Earth Syst. Sci. Discuss.*

2nd revision of Manuscript Number:  doi:10.5194/hess-2016-508, 2016

Title:  A Hydrological Prediction System based on the SVS Land-Surface Scheme: Implementation and Evaluation of the GEM-Hydro platform on the watershed of Lake Ontario

**General comments**

This is my second review of this paper which describes the development of a modelling system to estimate net basin supply to a strategic Canada/USA water body:  Lake Ontario.  In the following Specific comments Section, I believe that the comments in italic were not considered since the authors did not provide a rebuttal.

In my first review there were three basic objectives:

(i)     ''propose a methodology for calibrating the distributed GEM-Hydro platform developed by ECCC in order to improve streamflow simulations for Lake Ontario, which we expect would ultimately propagate into improved simulations of Lake Ontario Net Basin Supplies (or NBS, the sum of lake tributary runoff, overlake precipitation, and overlake evaporation: Brinkmann 1983);

(ii)    compare GEM-Hydro with two other distributed models (inter-comparison study) in order to identify avenues to further improve GEM-Hydro; and

(iii)   propose and evaluate a method for estimating runoff for the ungauged parts of the watershed.''

In this new version, I still find three stated objectives:

(i)     P.2, lines 16-17: « One of this paper's objectives is to present the first evaluation of the capabilities of the new SVS scheme for hydrological prediction in Canada

(ii)    P.4, line 5-7:  « …this study mainly aims at finding a methodology to implement the distributed GEM-Hydro model over the whole Lake Ontario watershed, including its ungauged parts, in an efficient manner. » - by the way, we do not find a methodology…we develop one – please consider replacing accordingly!

(iii)   P.4, lines 10-11: A second objective is to compare GEM-Hydro with two other distributed models (which is this study's contribution to GRIP-O) in order to identify avenues to further improve GEM-Hydro. »

It seems the new objective (ii) is a combination of the previous objectives (i) and (iii) minus evaluation of NBS with respect to those currently available.  This answers one of my previous comments.

As it is the paper reads more like a technical report than a scientific paper. I would have preferred a scientific paper that provides more fundamental information between the computational time scales of the LSS and those of WATROUTE and the UH. *For example, d discuss the relationship between the computational time scales and the dimension of the computational elements used in WATROUTE and the UH versus those used in the LSS.*

I encourage the authors to consider the comments introduced in my review as I feel the paper represents a good technological contribution to the hydrometeorological community.

**Specific comments**

- *P.1. consider modifying the end of the title as follows: …on the Lake Ontario basin, Canada*
  - *Please be consistent and use only one nomenclature, replace « watershed » by « basin » throughout the manuscript. Do not alternatively use catchment, basin and drainage area as synonymous. I know they are, but in a scientific document, it more conventional to use only terminology and since basin is used more than once in the manuscript, stick to it.*
- *P.4, Section 1.5: please correct me if I am wrong, but WATFLOOD has no LSS, just a simple potential evapotranspiration equation, unless WATCLASS was used. So WATFLOOD is more along the line of GR4J with that respect.*
- The following comment introduced in my first review should be addressed : *I think there is room here to provide more fundamental information between the computational time scales of the LSS and those of WATROUTE and UH. Furthermore, discuss the relationship between the computational time scales and the dimension of the computational elements used in WATROUTE and the UH.*
- P.5, line 34 and P.6, lines and 4, please be consistent use either « computation time » or « computational time », not both.
- *P.8, equation (1): why presenting the PBIAS expression and not the NS...the latter being more complex than the former…*
- P.11, lines 3-6*: the equifinality problem still exists for the global calibration, please discuss?* Despite global calibration is not be exempt of equifinality, the attention paid to the parameter ranges used (Table 3) allows to be confident in the physical relevance of the final parameter values.
- P. 12, line 20: Please Moira river should be written as Moira River throughout the manuscript, please be consistent
- P.12, line 34: the acronyms for Nash-Sutcliffe Coefficient using logarithmic values of streamflows are here NSE and NSE Ln, however, in some of the tables and Figures, the following acronyms Nash and Nash Ln are used, please use only one set of acronyms throughout the manuscript, not two.
- A similar comment applies for Pbias and PBIAS throughout the manuscript.

- P.13, lines 1 and 20 and in some of the tables and figures: the acronym for Nash-Sutcliffe Coefficient using square root values of streamflows are here Nash √and NSE √, please be consistent and use only one acronym throughout the manuscript, not two.
- P.17, lines 16-17: « However, as a limited number of subbasins were used for the inter-comparison due to computational time limitations, no general model ranking can be derived from this study. ».
  - *This means perhaps this paper is premature. Or as mentioned in the general comment section. Model intercomparison should be considered as supplemental information.*
- *I still do not get it, perhaps WATROUTE needs to be calibrated separately otherwise why calibrating with the UH? It is only valid to use WATROUTE if it can reproduce the UH at the chosen outlets used for the UH calibration. Unless there is a philosophical point I am not getting, which is perhaps possible, but doubtful. Please make a strong rebuttal to this statement.*

**Figures**

- There are two « Figure 3 », hence the second Figure 3 should be Figure 4, and Figures 4 and 5 should be Figures 5 and 6, respectively.
  - Replace sub-catchment by sub-basin, please be consistent.

**Tables**

- Tables3, 4 and 5
  - The range for some parameter values defies the imagination, any explanations?

**Answer to traditional questions**

Is the paper free of errors in logic?

- Yes

Do the conclusions follow from the evidence?

- Yes.

Are alternative explanations explored as appropriate?

- Yes.

Are biases, limitations, and assumptions clearly stated, and uncertainty quantified?

- Yes.

Is methodology explained in sufficient detail so that the paper's scientific conclusions could be tested by others?

- I am not sure

Is previous work and current understanding cited and represented correctly?

- Yes.

Is information conveyed clearly enough to be understood by the typical reader?

- Yes and no – I still have some minor issue related to the local and global calibration strategy

Are all figures and tables necessary, appropriate, legible, and annotated (as appropriate)?

- Yes.

---

## Author Response (AR2)

**Reviewer 1:**

Dear authors,
Thank you for considering the comments and revisions made. In my opinion, the revisions improved the clarity of presentations as well as context of the study. I believe, this might be an interesting contribution, worth to be published, however in its current form, I'm still not happy with the formulation of scientific objectives. The "implementation" (or describing the implementation) reads like the aim of some technical report, not a scientific investigation. Also the take home message is still not fully clear to me, meaning mainly some formulation of more general findings transferable to other regions/applications. What are the findings which might be potentially interesting for readers from other parts of the world? I understand that the study and results provide important local findings, but what has been learned in general? Please consider to discuss these aspects more clearly (perhaps consider to separate results and discussion section).

**Answer to reviewer (AR1):**

-In the Revision, the objectives are removed from the Introduction. Instead, we add the arguments about how this study is framed:

[revised manuscript text omitted]

Specific comments
1) Please consider to be more specific with definition of the objectives. " to present the first evaluation of the capabilities of the new SVS scheme for hydrological prediction"...hydrologic prediction of what?

AR2: See AR1

2) Avoid to cite unpublished manuscripts (Durnford et al. (in preparation))

AR3: the status of this citation was changed into "in Press"; the manuscript will probably be published as the author only received minor revisions (personal communication).

3) Please check Figure numbering.

AR4: Corrected.

4) Figure 9. What is the meaning of NBS.

AR5: This is clarified in the Revision.

**Reviewer 2**

[1]
*Hydrol. Earth Syst. Sci. Discuss.*
2nd revision of Manuscript Number: doi:10.5194/hess-2016-508, 2016
Title: A Hydrological Prediction System based on the SVS Land-Surface Scheme: Implementation and Evaluation of the GEM-Hydro platform on the watershed of Lake Ontario
**General comments**
This is my second review of this paper which describes the development of a modelling system to estimate net basin supply to a strategic Canada/USA water body: Lake Ontario. In the following Specific comments Section, I believe that the comments in italic were not considered since the authors did not provide a rebuttal.
In my first review there were three basic objectives:
(i) ''propose a methodology for calibrating the distributed GEM-Hydro platform developed by ECCC in order to improve streamflow simulations for Lake Ontario, which we expect would ultimately propagate into improved simulations of Lake Ontario Net Basin Supplies (or NBS, the sum of lake tributary runoff, overlake precipitation, and overlake evaporation: Brinkmann 1983);
(ii) compare GEM-Hydro with two other distributed models (inter-comparison study) in order to identify avenues to further improve GEM-Hydro; and
(iii) propose and evaluate a method for estimating runoff for the ungauged parts of the watershed.''

In this new version, I still find three stated objectives:
(i) P.2, lines 16-17: « One of this paper's objectives is to present the first evaluation of the capabilities of the new SVS scheme for hydrological prediction in Canada
(ii) P.4, line 5-7: « …this study mainly aims at finding a methodology to implement the distributed GEM-Hydro model over the whole Lake Ontario watershed, including its ungauged parts, in an efficient manner. » - by the way, we do not find a methodology…we develop one – please consider replacing accordingly!
AR6: done

(iii) P.4, lines 10-11: A second objective is to compare GEM-Hydro with two other distributed models (which is this study's contribution to GRIP-O) in order to identify avenues to further improve GEM-Hydro. »

It seems the new objective (ii) is a combination of the previous objectives (i) and (iii) minus evaluation of NBS with respect to those currently available. This answers one of my previous comments.

AR7: In the Revision, the objectives are removed from the Introduction. Instead, we add the arguments about how this study is framed.  See AR1.

As it is the paper reads more like a technical report than a scientific paper. I would have preferred a scientific paper that provides more fundamental information between the computational time scales of the LSS and those of WATROUTE and the UH. *For example, d discuss the relationship between the computational time scales and the dimension of the computational elements used in WATROUTE and the UH versus those used in the LSS.*

*AR8: a figure was added in this regard, and the text in the end of the models section was modified into this:*

" Figure 1 gives an overview of the relationship between computation time of the different models and the dimension of their domain. Note that GEM-Surf (Land-Surface part of GEM-Hydro) was run on ECCC's supercomputer while GEM-Hydro-UH and WATROUTE were run on a machine with an AMD Athlon Dual Core Processor 4800+, because GEM-Hydro-UH and WATROUTE are not parallelized yet (their computation time would not change substantially if run on ECCC's supercomputer). The computation time for the experiment setup described here and when splitting the domain in four on an ECCC supercomputer is about 1.5 min per day for GEM-Surf, provided that the pre-processing of the atmospheric variables was already done (which is the case in calibration: the pre-processing is done only once). WATROUTE (i.e., the routing part of GEM-Hydro) requires 25s per day for the setup described here when running on a local machine. The WATROUTE pre-processing (i.e., preparation of the WATROUTE input files from the SVS outputs, which would need to be done for each new run in calibration) takes about 30s per day and is quite constant whatever the domain size of the inputs fields. One simulation run over the GRIP-O period (4.5 years) therefore currently requires about 2 days with GEM-Hydro and prevents from performing any automatic calibration (which requires at least 400 runs, see below). GEM-Hydro-UH, based on a stand-alone version of SVS, saves a tremendous amount of computation time compared to GEM-Hydro mainly because of the Input/Output processing time: the stand-alone version makes use of text files which are kept open during the simulation and requires only 3s per day on a local machine for this setup (1.2 h for the 4.5 years GRIP-O period or 20 days of calibration with 400 runs if running the whole domain). However, the computation time required by WATROUTE still had to be bypassed to perform automatic calibrations, which was done with the UH concept. The UH (see for example Sherman, 1932) allows the estimation of the streamflow at the basin outlet by partitioning the basin averages of runoff and recharge in time. The same WATROUTE LZS formulation is used in GEM-Hydro-UH in order to estimate stream recharge. The basin averages required for the UH are computed as a weighted average of the SVS grid cells located in the considered basin. The UH only requires a decay parameter corresponding to the lag or response time of the considered basin, which controls the delay between the rainfall event and the resulting streamflow peak. It is estimated with the Epsey method (Almeida et al. 2014), which requires the basin area, perimeter, and the maximum and minimum elevations along the basin main river. The UH lag-time is also used as a free parameter during calibration (Table 2). It is inspired from the UH applied to the routing storage of GR4J (Perrin et al., 2003), but is employed here at an hourly time-step. Finally, this framework allows a considerable reduction of computation time and therefore allows to perform calibration. However, GEM-Hydro-UH is faster than GEM-Hydro as long as the domain size remains of the order of a few thousand points (see Figure 1). Beyond that threshold, not only calibration is not feasible any more with GEM-Hydro-UH, but it is possible that it becomes even slower than GEM-Hydro since the latter can be parallelized"

I encourage the authors to consider the comments introduced in my review as I feel the paper represents a good technological contribution to the hydrometeorological community.

**Specific comments**

• *P.1. consider modifying the end of the title as follows: ...on the Lake Ontario basin, Canada • Please be consistent and use only one nomenclature, replace « watershed » by « basin » throughout the manuscript. Do not alternatively use catchment, basin and drainage area as synonymous. I know they are, but in a scientific document, it more conventional to use only terminology and since basin is used more than once in the manuscript, stick to it.*

AR9: done

• *P.4, Section 1.5: please correct me if I am wrong, but WATFLOOD has no LSS, just a simple potential evapotranspiration equation, unless WATCLASS was used. So WATFLOOD is more along the line of GR4J with that respect.*

AR10: Indeed, the surface part of WATFLOOD cannot be considered a LSS because the model only solves the water balance instead of both the water and energy balances computed in a LSS. Therefore, first paragraph of section 1.1 was modified into the following (see mainly last sentence):

" Three different platforms are compared in this study: MESH, WATFLOOD, and GEM-Hydro. MESH and GEM-Hydro have in common a distributed representation of most hydrological processes occurring in a basin and a structure organized around two main components: a LSS for the representation of surface processes (evapotranspiration, infiltration, snow processes, water circulation in the soils), and a river routing scheme for simulating water transport in the streams, which consists of WATROUTE for all models. WATROUTE is a 1-D hydrologic routing model relying mainly on flow directions and elevation data (Kouwen 2010). It routes to the basin outlet the surface runoff and recharge produced by the surface schemes. In WATROUTE, runoff directly feeds the streams while recharge can be provided to an optional Lower Zone Storage (LZS) compartment, representing superficial aquifers, which releases water to the streams. WATFLOOD and GEM-Hydro make use of the LZS, whereas recharge from MESH feeds directly into the stream. WATFLOOD is not considered to include a LSS because it is not solving the energy balance, only the water balance, but it is distributed."

• *The following comment introduced in my first review should be addressed : I think there is room here to provide more fundamental information between the computational time scales of the LSS and those of WATROUTE and UH. Furthermore, discuss the relationship between the computational time scales and the dimension of the computational elements used in WATROUTE and the UH.*

AR11: see AR8.

• *P.5, line 34 and P.6, lines and 4, please be consistent use either « computation time » or « computational time », not both.*

AR12: done

• *P.8, equation (1): why presenting the PBIAS expression and not the NS...the latter being more complex than the former…*

AR13: done

• P.11, lines 3-6: *the equifinality problem still exists for the global calibration, please discuss?* Despite global calibration is not be exempt of equifinality, the attention paid to the parameter ranges used (Table 3) allows to be confident in the physical relevance of the final parameter values.

AR14: this sentence was added in section 1.4:Strategy for ungauged areas: " Global calibration is not exempt of equifinality issues either, but to a lower degree than local calibration. Indeed, the use of global parameters constrains parameter values across the basin to be equal and thus provides less freedom to achieve the same overall performance with different parameter sets. Moreover, the attention paid to the parameter ranges used (Table 2) allows to be confident in the physical relevance of the final parameter values."

• P. 12, line 20: Please Moira river should be written as Moira River throughout the manuscript, please be consistent

AR15: done

• P.12, line 34: the acronyms for Nash-Sutcliffe Coefficient using logarithmic values of streamflows are here NSE and NSE Ln, however, in some of the tables and Figures, the following acronyms Nash and Nash Ln are used, please use only one set of acronyms throughout the manuscript, not two.

AR16: done

• A similar comment applies for Pbias and PBIAS throughout the manuscript.

AR17: done

• P.13, lines 1 and 20 and in some of the tables and figures: the acronym for Nash-Sutcliffe Coefficient using square root values of streamflows are here Nash √and NSE √, please be consistent and use only one acronym throughout the manuscript, not two.

AR18: done

• P.17, lines 16-17: « However, as a limited number of subbasins were used for the inter-comparison due to computational time limitations, no general model ranking can be derived from this study. ». • *This means perhaps this paper is premature. Or as mentioned in the general comment section. Model intercomparison should be considered as supplemental information.*

AR19: done; most of the details and results related to the intercomparison has been moved to supplementary material; only a brief model description as well as the calibrated hydrographs for the Moira River are still in the main document, in order to support the fact that SVS may benefit from an improvement of winter-related processes, which CLASS (of MESH) seems to better represent.

• *I still do not get it, perhaps WATROUTE needs to be calibrated separately otherwise why calibrating with the UH? It is only valid to use WATROUTE if it can reproduce the UH at the chosen outlets used for the UH calibration. Unless there is a philosophical point I am not getting, which is perhaps possible, but doubtful. Please make a strong rebuttal to this statement.*

AR20: Yes WATROUTE needs to be calibrated separately when the calibration has been made with the UH; but this can be done manually in a few runs, as already stated in the manuscript in the conclusion: " The routing component of GEM-Hydro can be run afterwards, and re-calibrated separately. " And yes WATROUTE can reproduce the simulations obtained with the UH as already demonstrated in the manuscript on Figure 2, and mentioned by the following sentence of section 1.4:" In GEM-Hydro, standard Manning coefficients were used in WATROUTE, while the lag-time of GEM-Hydro-UH was adjusted during calibration. But it was assessed that simulations with GEM-Hydro (calibrated SVS and LZS parameters and standard Manning values) were very close, both in

terms of hydrographs and performances at the gauged sites, to those from the calibrated GEM-Hydro-UH. Performances are generally even slightly better with GEM-Hydro (despite the standard Manning values) than with GEM-Hydro-UH for individual subbasins (not shown), despite the opposite is true when looking at the total GRIP-O gauged area as a whole (see Table 5"; however, it is true that the performances of GEM-Hydro are quite significantly lower than those of GEM-Hydro-UH in Table 5 (formerly Table 8). I understand the concern of the reviewer in this regard. But this does not mean that GEM-Hydro cannot reach the same performances than GEM-Hydro-UH in this case, only that we didn't do enough manual tuning to achieve this. Therefore, a sentence was added in section 2.3 Runoff estimation for the whole Lake Ontario basin to clarify this:

" WATROUTE coefficients could have been manually tuned in order for GEM-Hydro performance values to reach those of GEM-Hydro-UH in Table 5, but this was not deemed necessary given the already very satisfying performance values obtained with the uncalibrated Manning values."

. Not reaching the exact same performances as GEM-Hydro-UH in Table 5 is not deemed more of an issue than this, because GEM-Hydro performances are still very good.

**Figures**
• There are two « Figure 3 », hence the second Figure 3 should be Figure 4, and Figures 4 and 5 should be Figures 5 and 6, respectively. o  Replace sub-catchment by sub-basin, please be consistent.
AR21: done

**Tables**
• Tables3, 4 and 5 o  The range for some parameter values defies the imagination, any explanations?
AR22: yes, we wanted to keep wide intervals to be sure not to constrain the model into non-optimal ranges, while keeping the parameter values realistic in terms of physical relevance. A sentence was added in the supplementary material to emphasize this:
" The parameter ranges of Tables 3-5 were generally chosen as wide as possible while remaining physically realistic, in order to let more freedom to the optimization algorithm, which may a priori increase the chances of finding optimal parameter sets during calibration."

**Answer to traditional questions**
Is the paper free of errors in logic?
• Yes

Do the conclusions follow from the evidence?
• Yes.

Are alternative explanations explored as appropriate?
• Yes.

Are biases, limitations, and assumptions clearly stated, and uncertainty quantified?
• Yes.

Is methodology explained in sufficient detail so that the paper's scientific conclusions could be tested by others?

• I am not sure

Is previous work and current understanding cited and represented correctly?
• Yes.

Is information conveyed clearly enough to be understood by the typical reader?
• Yes and no – I still have some minor issue related to the local and global calibration strategy

Are all figures and tables necessary, appropriate, legible, and annotated (as appropriate)?
• Yes.

[revised manuscript text omitted]

---

## Author Response (AR3)

[revised manuscript text omitted]

